# Temporal Misinformation and Conversion through Probabilistic Spiking Neurons

## Abstract

In the context of increasingly large neural network models and their associated high energy consumption, Spiking Neural Networks (SNNs) present a compelling alternative to Artificial Neural Networks (ANNs) due to their energy efficiency and closer alignment with biological neural principles. However, directly training SNNs with spatio-temporal backpropagation remains challenging due to their discrete signal processing and temporal dynamics. Alternative methods, notably ANN-SNN conversion, have enabled SNNs to achieve performance in various machine learning tasks, comparable to ANNs, but often to the expense of long latency needed to achieve such performance, especially on large scale complex datasets. The present work deals with ANN-SNN setting and identifies a new phenomenon we term "temporal misinformation", where random spike rearrangement through time in the converted SNN model improves its performance. To account for this, we propose bio-plausible, two-phase probabilistic (TPP) spiking neurons to be used in ANN-SNN conversion. We showcase the benefits of our proposed methods both theoretically and empirically through extensive experiments on CIFAR-10/100 and a large-scale dataset ImageNet over a variety of architectures, reaching SOTA performance.

## 1 Introduction

Spiking neural networks (SNNs), often referred to as the third generation of neural networks Maass (1997), are inspired by and designed to mimic how biological neurons process and share information McCulloch & Pitts (1943); Hodgkin & Huxley (1952); Izhikevich (2003). The efficiency of biological brains in terms of both energy use and task performance has long inspired the development of neural networks with similar capabilities. This inspiration has driven the growing interest in SNNs, particularly in time when large machine learning models demand increasingly high energy consumption. The main difference from the artificial neural networks (ANNs) Braspenning et al. (1995) comes from the way spiking neurons in an SNN process information. Spiking neurons communicate through a series of (discrete, often binary) spikes, emulating the biological brain's communication via electrical pulses. The (weighted) incoming spikes are accumulated in the neuron's membrane potential, and a spike is emitted only when the potential reaches

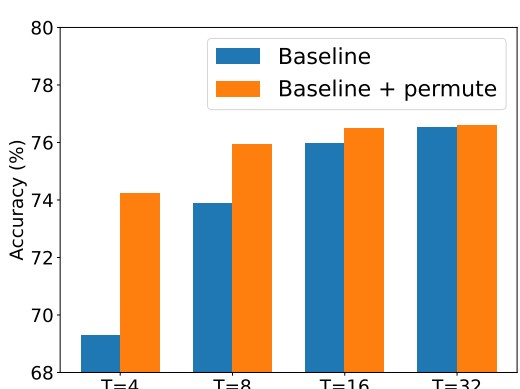

Figure 1: The initial experiment: After ANN-SNN conversion, we compared the accuracy of the baseline model with its "permuted" version, i.e. the baseline model but the output spike trains are permuted after each layer (setting is VGG16 - CIFAR100, ANN acc. 76.23%).

a threshold. This makes SNN processing event-driven and binary, and multiplication, as an energy demanding operation, is eliminated from the process. In contrast, ANNs process information using floating-point operations, which rely on multiplication, leading to energy-inefficient deep learning models at large scales Roy et al. (2019).

Furthermore, recent advancements in neuromorphic chip production Pei et al. (2019); DeBole et al. (2019); loi; Ma et al. (2023) further emphasize the advantages of SNN models. These chips, specifically designed to support and embed SNN models in hardware aware and efficient way, have opened new aspects of interest in SNNs, and various SNN models have been challenging traditional neural networks in various domains, including object detection Kim et al. (2020b); Cheng et al. (2020), object tracking Yang et al. (2019), video reconstruction Zhu et al. (2022), event camera and point clouds Ren et al. (2024), speech recognition Wang et al. (2023a) and generative models Kamata et al. (2022) such as SpikingBERT Bal & Sengupta (2024) and SpikeGPT Zhu et al. (2023); Wang et al. (2023b), to name a few.

Training SNNs presents a challenge in itself, due to the very same reasons from which the advantages of SNNs stem: their discrete processing of information. Unsupervised direct training, inspired by biological learning mechanisms, leverages local learning rules and spike timing to update weights Diehl & Cook (2015). While these methods are computationally friendly and could be performed on the specialized hardware, SNNs trained this way often underperform compared to models trained with other methods, and there is still plenty of room for the understanding and improvement of this method.

On the other side, supervised training methods can be categorized in two branches: direct training and ANN-SNN conversion based methods. The main challenge for direct training methods lies in the discrete nature of spike production. Namely, the operation of comparison of the membrane potential with the threshold is not differentiable, or, where it is, does not produce useful gradients. The success of direct training hinges on the development of spatio-temporal backpropagation through time (BPTT) and surrogate gradient methods O'Connor et al. (2018); Zenke & Ganguli (2018); Wu et al. (2018); Bellec et al. (2018); Fang et al. (2021a;b); Zenke & Vogels (2021); Mukhoty et al. (2024). Although, they address and overcome the main problem of non-differentiability of spikes, these methods encounter further challenges with deep architectures due to gradient instability and high computational costs during training simulations. Direct training focuses on optimizing not only synaptic weights but also dynamic parameters like firing thresholds Wei et al. (2023) and leaky factors Rathi & Roy (2023). Novel loss functions such as rate-based counting loss Zhu et al. (2024) and distribution-based loss Guo et al. (2022) were proposed to provide sufficient positive gradients and rectify the distribution of membrane potential during the propagation of binary spikes. Furthermore, hybrid training methods Wang et al. (2022b) combine ANN-SNN conversion with BPTT to achieve higher performance with low latency. Recent advancements include Ternary Spike Guo et al. (2024) for enhanced information capacity and the reversible SNN Zhang & Zhang (2024) to reduce memory costs during training.

The ground idea of ANN-SNN conversion is to use pre-trained ANN models to train an SNN. This starts by copying the weights of the ANN model to the SNN model following the same architecture, and then initializing hyperparameters of the spiking neurons in the SNN layers in such a way that the rate of the spikes approximate the values of the corresponding activation layers in the ANN. The advantages of this method lie in the fact that there is (usually) no extra computation needed for training the SNNs, so the computation of gradients can largely be avoided, or just reduced to the calculations during fine-tuning of the SNN model. This method (of which we will say more in Section 2) has been behind many of the state of the art performing SNNs, particularly on classification tasks.

The present work explores ideas that belong to the ANN-SNN conversion line of research. We start by identifying a phenomenon, that is rather counter-intuitive and, to the best of our knowledge, has gone unnoticed until now. Namely, when performing ANN-SNN conversion, the main assumption is that the sole carrier of information is the rate of the spiking activity, and precise timing of the spikes should not affect the performance of the SNN Bu et al. (2023). We challenged this assumption by using a baseline SNN obtained through ANN-SNN conversion, following methods proposed in recent literature. Then, when passing the samples to the baseline model, after each spiking layer we permuted the spike trains by rearranging the spikes in the temporal dimension. Specifically, the temporal order of spikes within each spike train was randomly shuffled. The permuted spike trains were then passed to the following layer in the SNN, and this process was continued until the output layer. The results of one of these initial experiments, comparing the performance of the "permuted" model with the original model, are presented in Figure 1. For every latency we performed this experiment, the "permuted" model surpassed the baseline and reached the original ANN accuracy much earlier. We dubbed this occurrence "the temporal misinformation" in ANN-SNN conversion and further explored it by giving it a more conceptual flavor in form of the bursting probabilistic spiking neurons which are designed to mimic the effect of permutations in SNNs. The proposed

neurons work in two-phases, in the first phase they collect the input (often beyond the threshold) while in the second they output spikes in a probabilistic manner with varying temporal probabilities. Two crucial properties that define our proposed spiking neurons, namely the accumulation of membrane potential beyond the threshold and entering into a firing phase (bursting), and probabilistic firing are bio-plausible, and were extensively studied in the neuroscience literature (see Section 3.3).

The main contributions of this paper are summarized as follows:

- We recognize the "temporal misinformation" phenomenon in ANN-SNN conversion, challenging the underlying assumption of ANN-SNN conversion which states that the spike rate is the sole carrier of information in the method.

- We propose a framework for its exploitation in ANN-SNN conversion utilizing two-phase probabilistic spiking neurons. We provide the theoretical insights into their functioning and superior performance, as well as support for their biological grounding.

- We performed a comprehensive experimental validation that demonstrates that our proposed method outperforms state-of-the-art conversion as well as the other training methods, in terms of accuracy on large scale CIFAR-10/100 and ImageNet datasets.

## 2 PRELIMINARIES

The base model that we use in this paper is Integrate-and-Fire (IF) spiking neuron whose internal dynamics, after discretization, is given by the equations

$$\mathbf{v}^{(l)}[t] = \mathbf{v}^{(l)}[t-1] + \mathbf{W}^{(l)}\theta^{(l-1)} \cdot \mathbf{s}^{(l-1)}[t] - \theta^{(l)} \cdot \mathbf{s}[t-1], \tag{1}$$

$$\mathbf{s}^{(l)}[t] = H(\mathbf{v}^{(l)}[t] - \theta^{(l)}). \tag{2}$$

Here, $\theta^{(l)}$ is the threshold (vector), $H(\cdot)$ is the Heaviside function, while the superscript $l$ pertains to the layer in the SNN. Later on, we will later modify these equations and use more advanced neuron models, but for now, by unrolling the equations through $t = 1, \ldots, T$, and rearranging the terms, we obtain

$$\theta^{(l)} \frac{\sum_{t=1}^{T} \mathbf{s}^{(l)}[t]}{T} = \mathbf{W}^{(l)} V_{\text{th}}^{(l-1)} \frac{\sum_{t=1}^{T} \mathbf{s}^{(l-1)}[t]}{T} \tag{3}$$

$$+ \frac{\mathbf{v}^{(l)}[T] - \mathbf{v}^{(l)}[0]}{T}. \tag{4}$$

On the ANN side, a passage between the layers takes the form

$$a^{(l)} = \mathcal{A}^{(l)}(\mathbf{W}^{(l)} a^{(l-1)}), \tag{5}$$

where $\mathcal{A}^{(l)}$ is the activation function. The ANN-SNN conversion process starts with copying the weights (and biases) of a pre-trained ANN model to the SNN model following the same architecture. Then, by comparing the equations for the ANN outputs equation 5 and the average output of the SNN equation 3 Rueckauer et al. (2017a), one ideally wants a relation of the form

$$a_i^{(l)} \approx V_{\text{th}}^{(l)} \frac{\sum_{t=1}^{T} \mathbf{s}_i^{(l)}[t]}{T}. \tag{6}$$

The most commonly used activation function $\mathcal{A}$ is ReLU , due to its simplicity and non-negative output, which aligns well with the properties of IF neurons.

For a successful conversion that leads to minimal conversion error, one can note the importance of the three components, namely: 1) The threshold value $\theta$, 2) The initialization $\mathbf{v}[0]$, 3) The ANN activation function $\mathcal{A}$.

### 2.1 RELATED WORK

**ANN-SNN conversion** leverages pre-trained ANNs to initialize SNNs, aiming to minimize accuracy degradation by aligning ANN activations with SNN firing rates, as demonstrated in early works Rueckauer et al. (2017a); Cao et al. (2015). Subsequent studies addressed conversion errors

and improved temporal accuracy through techniques like weight normalization Diehl et al. (2015), soft-reset mechanisms Rueckauer et al. (2017b); Han et al. (2020), and dynamic threshold adjustment Stöckl & Maass (2021); Ho & Chang (2021); Wu et al. (2023). Efficient conversion with fewer spikes was achieved through rate-coding and time-coding methods Kim et al. (2020a), as well as specialized weight renormalization Sengupta et al. (2018).

A recent direction involves modifying the ANN activation functions to reduce conversion errors. Methods using thresholded ReLU activation Ding et al. (2021) and quantized activation functions Bu et al. (2022c); Liu et al. (2022); Hu et al. (2023); Shen et al. (2024) have achieved high accuracy at lower latencies. However, these approaches often reduce the original ANN accuracy, limiting the potential performance of the converted SNN. Techniques like Li & Zeng (2022); Wang et al. (2022a); Liu et al. (2022) propose modifications to the inner function of IF neurons to reduce conversion errors. Notably, a two-phase spiking neuron mechanism similar to ours has been used in Liu et al. (2022).

Membrane potential and threshold initialization play crucial roles in reducing conversion errors. Many methods utilize layer-wise maximum ANN activations, or some percentile of them, for threshold initialization Rueckauer et al. (2017a); Deng & Gu (2021a); Li et al. (2021). Detailed studies on membrane potential initialization and threshold settings are provided in Hao et al. (2023a); Bojkovic et al. (2024). Post-conversion weight calibration Li et al. (2021); Bojkovic et al. (2024) further enhances SNN performance, leading to hybrid training methods that combine ANN-SNN conversion with fine-tuning.

In general, one can argue that ANN-SNN conversion based methods of training SNNs can be classified in two categories. The first line of thought deals with modification on the ANN side, most notably in quantization of the ANN activation functions, in order to reduce the conversion error in low latency. The second line deals with modification on ANN side, where the spiking neuron mechanisms are modified in order to reduce this error. The advantage in the former case comes from the lower latency to have a good performance, but the disadvantage comes from the fact that quantization of the ANN activations in general, yields the poorer ANN performance, hence limits the SNN performance as well. In the latter case, the situation is reversed, the ANNs utilized have higher performance, but SNNs sometimes need longer latency to achieve it. Our approach belongs to the second category.

**Direct training** allows SNNs to exploit precise spike timing and operate within a few timesteps. The success of direct training hinges on the development of spatio-temporal backpropagation through time (BPTT) and surrogate gradient methods O'Connor et al. (2018); Zenke & Ganguli (2018); Wu et al. (2018); Bellec et al. (2018); Fang et al. (2021a;b); Zenke & Vogels (2021); Mukhoty et al. (2024). However, these methods encounter challenges with deep architectures due to gradient instability and high computational costs during training simulations. Various gradient-based methods leverage surrogate gradients O'Connor et al. (2018); Zenke & Ganguli (2018); Wu et al. (2018); Bellec et al. (2018); Fang et al. (2021a;b); Zenke & Vogels (2021); Mukhoty et al. (2024) to address the non-differentiable nature of spike functions. Direct training focuses on optimizing not only synaptic weights but also dynamic parameters like firing thresholds Wei et al. (2023) and leaky factors Rathi & Roy (2023). Novel loss functions such as rate-based counting loss Zhu et al. (2024) and distribution-based loss Guo et al. (2022) were proposed to provide sufficient positive gradients and rectify the distribution of membrane potential during the propagation of binary spikes. Furthermore, hybrid training methods Wang et al. (2022b) combine ANN-SNN conversion with BPTT to achieve higher performance with low latency. Recent advancements include Ternary Spike Guo et al. (2024) for enhanced information capacity and the reversible SNN Zhang & Zhang (2024) to reduce memory costs during training.

## 3 MOTIVATION AND PROPOSED METHOD

When performing ANN-SNN conversion, one usually employs constant or rate encoding in the obtained SNN model, with the underlying idea that the expectation of the input at each time step is equal to the original input to the ANN model. In particular, there is no temporal information in the encoding, as the precise timing of spikes does not carry any extra information. In the constant encoding this is obvious, while in the rate encoding, for a fixed input channel, and for every time step, the probability of having the spike is constant (and equal to the value of the channel assumed to be between 0 and 1).

The obtained SNN model is initialized in such a way that it approximates the outputs of the starting ANN model, through the paradigm that for each spiking neuron, the average number of spikes it produces, or its expectation of the output, should approximate the output of the corresponding ANN neuron. In particular, one assumes and expects that there is no temporal information throughout the SNN model, i.e. the spike train outputs of each SNN layer should not carry any extra temporal information, other than the spike firing rates.

To our surprise, we discovered that this was not the case (see Figure 1).

### 3.1 Permuting spike trains

To test the initial hypothesis of the absence of "temporal information", we designed an experiment where for an SNN model obtained through the ANN-SNN conversion, after each layer we would collect the output spike trains, and permute them through the temporal dimension. More precisely, for a fixed latency $T$ and for each spiking layer, we would collect the output spike trains of temporal length $T$, permute them, and pass them to the next layer, and continue this process until the output layer. We used the constant encoding for the input. We further compared the performance of this model with the original base SNN model, whose output spike trains have not been manipulated through permutations.

The performance of the base and "permuted" SNN models has been compared in two ways. First, for the latency $T$ and for the latencies $t < T$. What we discovered is that if we consider the latency $T_{top}$ where the base model achieves the top accuracy, the performance of the two models is pretty much the same. However, if we consider the latency $T < T_{top}$, the "permuted" model outperforms the base model, in some cases drastically. Moreover, the situation becomes more contrasted if we consider the latencies $t < T$. The reader can refer to the Figure 1 for more information, while the details of the experiment are in the Appendix.

The conclusion of these initial experiments is that, contrary to the expectation, ANN-SNN conversion is not invariant under the temporal manipulation of output spike trains. Moreover, the effect of permuting the spike trains yields better performance of the converted SNN model, a phenomenon to which we refer as **temporal misinformation** in ANN-SNN conversion.

### 3.2 From permutations to Bursting Probabilistic Spiking Neurons

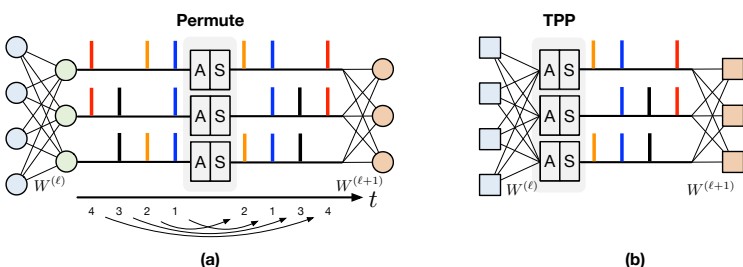

Figure 2: (a) The "permutation" layer collects the spike outputs of the layer in the first **A**ccumulation phase, while in the second **S**piking phase it outputs the same number of spikes as the original spike train, but with permuted firing times. (b) Bursting probabilistic spiking neurons accumulate the weighted outputs from the previous layer and then output them according to their inner dynamics.

The previous sections hint at the motivation of the present work. Our aim is to answer the question: How to incorporate the action of permutation of the output spike trains into the dynamics of the spiking neurons?

Suppose that we want to permute the spikes trains coming from the layer $\ell$. A general idea would be to have a "permutator"- a layer immediately after, whose goal would be to collect all the spikes, and outputs them in a permuted fashion, and sends such obtained spike trains to the following layer. One may refer to Figure 2 (a) for the visual representation of this concept. This immediately suggests the **two-phase** nature of the "permutator", namely, in the first phase the incoming spikes are accumulated and the firing is delayed until the beginning of the second, firing phase.

The second line of thought concerns the nature of the outputting mechanism of the "permutator". In particular, we would like to have a mechanism of spiking neurons which keep the "random" component of the permutations. This lead us to the probabilistic firing of spiking neurons.

The final question that we consider is, can we make the situation more compact, by using probabilistic spiking neuron which would collect the weighted input of the previous layer (rather than the spikes of the spiking layer), and output what would be "permutation" of spike trains (see Figure 2 (b))?

**TPP neurons** The answer to all of the above is given in form of the proposed **two-phase probabilistic spiking neurons** (TPP). Namely, in the first phase, the neurons will only accumulate the (weighted) input coming from the previous layer, while in the second phase, the neurons will spike. More precisely, suppose that at a particular layer $\ell$ the spiking neurons accumulate the whole output of the previous layer, without emitting spikes. Let us denote the accumulated membrane potential by $\mathbf{v}^{(l)}[0]$. Then, the spiking phase is described with equations

$$
\mathbf{s}^{(l)}[t] = B \left( \frac{1}{\theta^{(l)} \cdot (T - t + 1)} \mathbf{v}^{(l)}[t - 1] \right),
$$
$$
\mathbf{v}^{(l)}[t] = \mathbf{v}^{(l)}[t - 1] - \theta^{(l)} \cdot \mathbf{s}[t],
$$
(7)

and $t = 1, \ldots, T$. Here, $B(x)$ is a Bernoulli random variable with bias $x$, extended for $x \in \mathbb{R}$ in a natural way ($B(x) = B(\max(\min(x, 1), 0))$). If the weights of the SNN network are not normalized, the produced spikes will be scaled with the thresholds $\theta^{(l)} \cdot \mathbf{s}^{(l)}[t]$, before being sent to the next layer.

One may notice that the presence of $T - t + 1$ in the denominator of the bias in $B$, implying that the probability of spiking does not only depend on the current membrane potential, but also on the time step: in the absence of spiking, for the same membrane potential, the probability of spiking increases through time.

**Total output** Although the proposed spiking activity is probabilistic, the total output of the spiking neuron (the number of spikes) expresses little variability, which is seen in the following.

**Theorem 1.** *Suppose that for some $0 < t < T$, we have $t \cdot \theta^{(l)} \leq \mathbf{v}^{(l)}[0] < (t + 1) \cdot \theta^{(l)}$, and we are in the setting of equation 7. Then, the probability that the neuron will spike more than $t + 1$ times, or less than $t$ times is zero. Moreover, the probability of having a spike at any given time step $t = 1, \ldots, T$ is non-zero.*

The proof is given in the Appendix, but we may note that the result states that TPP neurons output the exact number of spikes as they should, and those spikes can have arbitrary positioning throughout the time steps. In other words, they act somewhat as a "permutation" on the output spike trains.

**Heuristics behind permutations** We come back to the original motivation, and the mysterious effect of temporal misinformation. To this end, we notice that permutations may act as a "uniformizer" of the inputs to the spiking neuron, which is highly related to notions of phase lag or unevenness of the inputs (see Li et al. (2022) and Bu et al. (2022c), respectively).

**Theorem 2.** *Suppose we have $N$ spiking neurons that produced spike trains $s_i[1], s_i[2], \ldots, s_i[T]$, $i = 1, \ldots, N$. Furthermore, suppose that these spike trains are modulated with weights $w_1, \ldots, w_N$, and as such give input to a neuron (say from the following layer) in the form $x[t] = \sum w_i s_i[t]$, for $t = 1, \ldots, T$. For a given permutation $\pi = (\pi_1, \ldots, \pi_N)$, let $\pi s_i$ denote the permutation of*

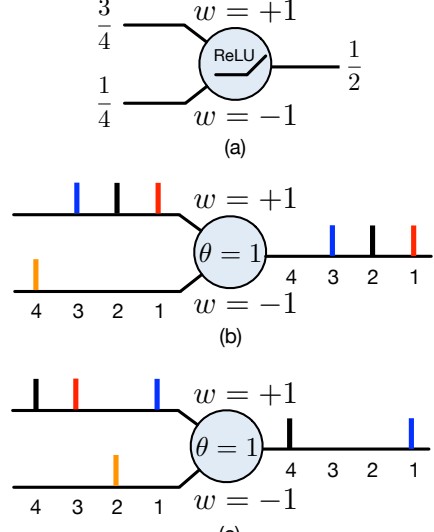

Figure 3: (a) ReLU activation with inputs of $\frac{3}{4}$ and $\frac{1}{4}$, and corresponding weights of $+1$ and $-1$. After summing, the ground truth output is $\frac{1}{2}$; (b) Baseline case: input spike trains without permutation yields an ANN-SNN conversion error $\frac{1}{4}$ due to delayed spike at $t = 4$ (orange spike); (c) Spike trains with permutation applied to move delayed spikes at $t = 4$ forward to $t = 2$. This adjustment heuristically aligns the output with the original ANN output $\frac{1}{2}$.

*the spike train $s_i$. Then, for every $t_1, t_2 \in \{1, 2, \ldots, T\}$,*

$$E_\pi[\sum w_i \pi s_i[t_1]] = E_\pi[\sum w_i \pi s_i[t_2]].$$

The previous result deals with the expected outputs with respect to the permutations. When it comes to the action of a single permutation, we make the following observation. The effect of a single permutation is mostly visible on spike trains that have a **low number of spikes**. This, in turn, is related to the situation where the input to the neuron is low throughout time, and it takes longer for a neuron to accumulate enough potential in order to spike, hence the neuron spikes at a later time during latency. In this case, a single permutation of the output spike(s) actually move the spikes forward in time (in general) and as such contributes to the elimination of the unevenness error, which appears when the input to a neuron in the beginning is higher than the average input through time (hence, the neuron produces superfluous spikes in the beginning, which shouldn't be the case), see Figure 3.

### 3.3 BIO-PLAUSIBILITY AND HARDWARE IMPLEMENTATION OF TPP NEURONS

Our proposed neurons have two distinct properties: The two-phase regime and probabilistic spike firing. Both of these properties are biologically plausible and extensively studied in the neuroscience literature. For example, the two phase regime can be related to firing after a delay of biological spiking neurons, where a neuron collects the input beyond the threshold value and fires after delay or after some condition is met. It could also be related to the bursting, when a biological neuron starts emitting bursts of spikes, after a certain condition is met, effectively dumping their accumulated potential. One can refer to Izhikevich (2007); Connors & Gutnick (1990); Llinás & Jahnsen (1982); Krahe & Gabbiani (2004) for more details.

On the other side, stochastic firing of biological neurons has been well studied as well, and different aspects of noise introduction into firing have been proposed. One can refer to Shadlen & Newsome (1994); Faisal et al. (2008); Softky & Koch (1993); Maass & Natschläger (1997); Pagliarini et al. (2019); Stein et al. (2005), for some examples.

When it comes to implementation of TPP neurons on neuromorphic hardware, two phase regime can be easily achieved on many of the modern neuromorphic that support programmable spiking neurons. The stochastic firing can be achieved through random sampling which is, for example, supported on IBM TrueNorth Merolla et al. (2014), Intel Loihi Davies et al. (2018), BrainScaleS-2 Pehle et al. (2022), SpiNNaker Furber et al. (2014) neuromorphic chips.

The probabilistic spiking mechanism we introduce aligns with the stochastic firing behaviors observed in biological neurons, a feature that has been effectively implemented in neuromorphic hardware such as IBM's TrueNorth DeBole et al. (2019); Merolla et al. (2014), Intel's Loihi loi; Davies et al. (2018), BrainScaleS-2 Pehle et al. (2022), SpiNNaker and SpiNNaker2. For example, TrueNorth incorporates stochastic neuron models using on-chip pseudo-random number generators, enabling probabilistic firing patterns that mirror our approach. Similarly, Loihi Gonzalez et al. (2024) supports stochastic operations by adding uniformly distributed pseudorandom noise to neuronal variables, facilitating the implementation of probabilistic spiking neurons.

To reduce the overall latency for processing inputs with our models, which yields linear dependence on the number of layers (implied by the two phase regime), we note that as soon as a particular layer has finished the firing phase, it can start receiving the input from the previous layer: The process of classifying a dataset can be serialized. This has already been observed, for example in Liu et al. (2022). Neuromophic hardware implementation of this serialization has been proposed as well, see for example Das (2023); Song et al. (2021); Varshika et al. (2022).

## 4 EXPERIMENTS

In this section, we verify the effectiveness and efficiency of our proposed methods. We compare it with state-of-the-art methods for image classification via converting ResNet-20, ResNet-34 He et al. (2016), VGG-16 Simonyan & Zisserman (2015), RegNet Radosavovic et al. (2020) on CIFAR-10 LeCun et al. (1998); Krizhevsky et al. (2010), CIFAR-100 Krizhevsky & Hinton (2009), and ImageNet Deng et al. (2009). Our experiments use PyTorch Paszke et al. (2019), PyTorch vision

models maintainers & contributors (2016), and the PyTorch Image Models (Timm) library Wightman (2019).[1]

To demonstrate the wide applicability of the TPP neurons and the framework we propose, we combine them with three representative methods of ANN-SNN conversion from recent literature, each of which has their own particularities. These methods are: QCFS Bu et al. (2022b), RTS Deng & Gu (2021a), and SNNC Li et al. (2021). The particularity of QCFS method is that it uses step function instead of ReLU in ANN models during their training, in order to obtain higher accuracy in lower latency after the conversion. RTS method uses thresholded ReLU activation in ANN models during their training, so that the outliers are eliminated among the activation values, which helps to reduce the conversion error. Finally, SNNC uses standard ANN models with ReLU activation, and performs grid search on the activation values to find optimal initialization of the thresholds in the converted SNNs.

We initialize our SNNs following the standard ANN-SNN conversion process described in Section 3 (and detailed in A), starting with a pre-trained model given by the baseline, or with training an ANN model using default settings in QCFS Bu et al. (2022b), RTS Deng & Gu (2021a), and SNNC Li et al. (2021). ANN ReLU activations were replaced with layers of TPP neurons initialized properly. All experiments were conducted using NVIDIA RTX 4090 and Tesla A100 GPUs. For comprehensive details on all setups and configurations, see Appendix C.2.

### 4.1 COMPARISON WITH THE STATE-OF-THE-ART ANN-SNN CONVERSION METHODS

We evaluate our approach against previous state-of-the-art ANN-SNN conversion methods, including ReLU-Threshold-Shift (RTS) Deng & Gu (2021a), SNN Calibration with Advanced Pipeline (SNNC-AP) Li et al. (2021), Quantization Clip-Floor-Shift activation function (QCFS) Bu et al. (2022b), SNM Wang et al. (2022a), Burst Li & Zeng (2022), OPI Bu et al. (2022a), SRP Hao et al. (2023a), DDI Bojkovic et al. (2024) and FTBC et al. (2024).

Table 1: Comparison between our method and the other ANN-SNN conversion methods on ImageNet. We provide the average accuracy and the associated standard deviation across 5 experiments (for our methods, we need extra $c$ steps for summation, see Section 3.2).

| Architecture | Method | ANN | T=4 | T=8 | T=16 | T=32 | T=64 | T=128 |
|---|---|---|---|---|---|---|---|---|
| ResNet-34 | RTS Deng & Gu (2021a) | 75.66 | – | – | – | 33.01 | 59.52 | 67.54 |
| | SNNC-AP*Li et al. (2021) | 75.66 | – | – | – | 64.54 | 71.12 | 73.45 |
| | QCFS Bu et al. (2022b) | 74.32 | – | – | 59.35 | 69.37 | 72.35 | 73.15 |
| | SRP Hao et al. (2023a) | 74.32 | 66.71 | 67.62 | 68.02 | 68.40 | 68.61 | – |
| | FTBC(+QCFS) et al. (2024) | 74.32 | 49.94 | 65.28 | 71.66 | 73.57 | 74.07 | 74.23 |
| | **Ours (TPP) + QCFS** | 74.32 | 37.23 (0.07) | 67.32 (0.06) | 72.03 (0.02) | 72.97 (0.03) | 73.24 (0.02) | 73.30 (0.02) |
| | **Ours (TPP)*+ SNNC w/o Cali.** | 75.65 | 2.69 (0.03) | 49.24 (0.23) | 69.97 (0.10) | 74.07 (0.06) | 75.23 (0.03) | 75.51 (0.05) |
| VGG-16 | SNNC-AP*Li et al. (2021) | 75.36 | – | – | – | 63.64 | 70.69 | 73.32 |
| | SNM*Wang et al. (2022a) | 73.18 | – | – | – | 64.78 | 71.50 | 72.86 |
| | RTS Deng & Gu (2021a) | 72.16 | – | – | 55.80 | 67.73 | 70.97 | 71.89 |
| | QCFS Bu et al. (2022b) | 74.29 | – | – | 50.97 | 68.47 | 72.85 | 73.97 |
| | Burst Li & Zeng (2022) | 74.27 | – | – | – | 70.61 | 73.32 | 73.00 |
| | OPI*Bu et al. (2022a) | 74.85 | – | 6.25 | 36.02 | 64.70 | 72.47 | 74.24 |
| | SRP Hao et al. (2023a) | 74.29 | 66.47 | 68.37 | 69.13 | 69.35 | 69.43 | – |
| | FTBC(+QCFS) et al. (2024) | 73.91 | 58.83 | 69.31 | 72.98 | 74.05 | 74.16 | 74.21 |
| | **Ours (TPP) + RTS** | 72.16 | 30.50 (1.19) | 56.69(0.67) | 67.34 (0.25) | 70.63 (0.11) | 71.75 (0.05) | 72.05 (0.03) |
| | **Ours (TPP) + QCFS** | 74.22 | 68.39 (0.08) | 72.99 (0.05) | 73.98 (0.07) | 74.23 (0.03) | 74.29 (0.00) | 74.33 (0.01) |
| | **Ours (TPP)*+ SNNC w/o Cali.** | 75.37 | 54.14 (0.59) | 69.75 (0.27) | 73.44 (0.02) | 74.72 (0.06) | 75.14 (0.02) | 75.25 (0.03) |
| RegNetX-4GF | RTS Deng & Gu (2021a) | 80.02 | – | – | – | 0.218 | 3.542 | 48.60 |
| | SNNC-AP*Li et al. (2021) | 80.02 | – | – | – | 55.70 | 70.96 | 75.78 |
| | **Ours (TPP)*+ SNNC w/o Cali.** | 78.45 | – | – | 22.71 (2.98) | 66.51 (0.44) | 75.54 (0.07) | 77.83 (0.04) |

\* Without modification to ReLU of ANNs.

**ImageNet dataset:** Table 1 compares the performance of our proposed methods with state-of-the-art ANN-SNN conversion methods on ImageNet. Our method outperforms the baselines across all simulation time steps for VGG-16, and RegNetX-4GF. For instance, on VGG-16 at $T = 32$, our method achieves 74.72% accuracy, surpassing other baselines even at $T = 128$. Moreover, at $T = 128$, our method nearly matches the original ANN performance with only a 0.12% drop in VGG-16 and a 0.14% drop in ResNet-34.

---

[1]https://github.com/huggingface/pytorch-image-models

We see similar patterns in combining our methods with RTS and QCFS baselines, which use modified ReLU activations to reduce conversion errors. Table 1 shows these results. For instance, applying TPP with QCFS on ResNet-34 at $T = 16$ improves performance from 59.35% to 72.03%, a 12.68% increase. Similarly, for VGG-16 at $T = 16$, combining TPP with QCFS boosts performance from 50.97% to 73.98%, a 23.01% increase. Using TPP with RTS also shows significant improvements, such as a 12.82% increase for VGG-16 at $T = 16$. These results demonstrate the benefits of integrating TPP with other optimization approaches, solidifying its role as a comprehensive solution for ANN-SNN conversion challenges.

Table 2: Comparison between our proposed method and other ANN-SNN conversion methods on CIFAR-100 dataset. The average accuracy and standard deviation of the TPP method are reported over 5 experiments (for our methods, we need extra $c$ steps for summation, see Section 3.2).

| Architecture | Method | ANN | T=4 | T=8 | T=16 | T=32 | T=64 | T=128 |
|---|---|---|---|---|---|---|---|---|
| ResNet-20 | TSC* Han & Roy (2020) | 68.72 | – | – | – | – | – | 58.42 |
| | RMP* Han et al. (2020) | 68.72 | – | – | – | 27.64 | 46.91 | 57.69 |
| | SNNC-AP* Li et al. (2021) | 77.16 | – | – | 76.32 | 77.29 | 77.73 | 77.63 |
| | RTS Deng & Gu (2021a) | 67.08 | – | – | 63.73 | 68.40 | 69.27 | 69.49 |
| | OPI* Bu et al. (2022a) | 70.43 | – | 23.09 | 52.34 | 67.18 | 69.96 | 70.51 |
| | QCFS* Bu et al. (2022b) | 67.09 | 27.87 | 49.53 | 63.61 | 67.04 | 67.87 | 67.86 |
| | Burst* Li & Zeng (2022) | 80.69 | – | – | – | 76.39 | 79.83 | 80.52 |
| | **Ours (TPP) + QCFS** | 67.10 | 46.88 (0.40) | 64.77 (0.20) | 67.25 (0.12) | 67.74 (0.06) | 67.77 (0.05) | 67.79 (0.04) |
| | **Ours (TPP)\*+ SNNC w/o Cali.** | 81.89 | 39.67 (0.99) | 71.05 (0.68) | 78.97 (0.24) | 81.06 (0.05) | 81.61 (0.08) | 81.62 (0.05) |
| VGG-16 | TSC* Han & Roy (2020) | 71.22 | – | – | – | – | – | 69.86 |
| | SNM* Wang et al. (2022a) | 74.13 | – | – | – | 71.80 | 73.69 | 73.95 |
| | SNNC-AP* Li et al. (2021) | 77.89 | – | – | – | 73.55 | 77.10 | 77.86 |
| | RTS* Deng & Gu (2021a) | 76.13 | 23.76 | 43.81 | 56.23 | 67.61 | 73.45 | 75.23 |
| | OPI* Bu et al. (2022a) | 76.31 | – | 60.49 | 70.72 | 74.82 | 75.97 | 76.25 |
| | QCFS* Bu et al. (2022b) | 76.21 | 69.29 | 73.89 | 75.98 | 76.53 | 76.54 | 76.60 |
| | DDI Bojkovic et al. (2024) | 70.44 | 51.21 | 53.65 | 57.12 | 61.61 | 70.44 | 73.82 |
| | FTBC(+QCFS) et al. (2024) | 76.21 | 71.47 | 75.12 | 76.22 | 76.48 | 76.48 | 76.48 |
| | **Ours (TPP) + RTS** | 76.13 | 37.88 (0.35) | 65.81 (0.27) | 73.05 (0.12) | 75.17 (0.17) | 75.64 (0.12) | 75.9 (0.08) |
| | **Ours (TPP) + QCFS** | 76.21 | 73.93 (0.22) | 76.03 (0.23) | 76.43 (0.07) | 76.55 (0.03) | 76.55 (0.07) | 76.52 (0.04) |
| | **Ours (TPP)\*+ SNNC w/o Cali.** | 77.87 | 59.23 (0.65) | 73.16 (0.17) | 76.05 (0.26) | 77.16 (0.09) | 77.56 (0.13) | 77.64 (0.04) |

\* Without modification to ReLU of ANNs.
+ Using authors' provided models and code.
° Self implemented.

**CIFAR dataset:** We further evaluate the performance of our methods on CIFAR-100 dataset and present the results in Table 2. We observe similar patterns as with the ImageNet. When comparing our method with ANN-SNN conversion methods which use non-ReLU activations, e.g. QCFS and RTS, our method constantly outperforms RTS on ResNet-20 and VGG16. QCFS baseline suffers from necessity to train ANN models from scratch with custom activations, while our method is applicable to any ANN model with ReLU -like activation. Furthermore, custom activation functions sometimes sacrifice the ANN performance as can be seen from the corresponding ANN accuracies.

## 4.2 COMPARISON WITH OTHER TYPES OF SNN TRAINING METHODS AND MODELS

We compare our approach with several state-of-the-art direct training and hybrid training methods as presented in Table 3. The comparison is founded on performance metrics like accuracy and the number of timesteps utilized during inference on the CIFAR-100 and ImageNet datasets. We benchmark our method against prominent approaches such as LM-H Hao et al. (2023b), SEENN Li et al. (2023), Dual-Phase Wang et al. (2022b), TTS Guo et al. (2024), RMP-Loss Guo et al. (2023), RecDis-SNN Guo et al. (2022), SpikeConv Liu et al. (2022), and GAC-SNN Qiu et al. (2024). We showcase the best accuracy comparable to state-of-the-art methods achieved by our approach with minimal timesteps. We prioritize accuracy, but direct training and hybrid training opt for a lower number of timesteps and sacrifice accuracy. We outperform LM-H Hao et al. (2023b) and Dual-Phase Wang et al. (2022b) for VGG-16 on CIFAR-100. For ResNet-20 on CIFAR-100, we have higher accuracy but longer timesteps. Additionally, for ResNet-34 on the ImageNet dataset, the accuracy of our method with QCFS with 16 timesteps is higher than that of SpikeConv Liu et al. (2022) with the same number of timesteps. We also achieve higher accuracy with longer timesteps as expected. Overall, our approach demonstrates promising performance and competitiveness in comparison with the existing SNN training methods.

Table 3: Comparison with direct and hybrid training methods for SNNs on CIFAR-100 and ImageNet datasets. For baselines, we report their highest reported accuracy and the corresponding latency.

| Dataset | Architecture | Method | Category | Timesteps | Accuracy |
|---|---|---|---|---|---|
| CIFAR-100 | VGG-16 | LM-H Hao et al. (2023b) | Hybrid Training | 4 | 73.11 |
| | | SEENN-II [*]Li et al. (2023) | Direct Training | 1.15[*] | 72.76 |
| | | Dual-Phase Wang et al. (2022b) | Hybrid Training | 4 / 8 | 70.08 / 75.06 |
| | | **Ours (TPP) + QCFS** | ANN-SNN | 4 / 8 | 73.93 / 76.03 |
| | ResNet-20 | LM-H Hao et al. (2023b) | Hybrid Training | 4 | 57.12 |
| | | TTS Guo et al. (2024) | Direct Training | 4 | 74.02 |
| | | **Ours (TPP) + SNNC w/o Cali.** | ANN-SNN | 16 | 78.97 |
| ImageNet | ResNet-34 | SEENN-I Li et al. (2023) | Direct Training | 3.38[*] | 64.66 |
| | | RMP-Loss Guo et al. (2023) | Direct Training | 4 | 65.17 |
| | | RecDis-SNN Guo et al. (2022) | Direct Training | 6 | 67.33 |
| | | SpikeConv Liu et al. (2022) | Hybrid Training | 16 | 70.57 |
| | | GAC-SNN Qiu et al. (2024) | Direct Training | 6 | 70.42 |
| | | TTS Guo et al. (2024) | Direct Training | 4 | 70.74 |
| | | SEENN-I Li et al. (2023) | Direct Training | 29.53[*] | 71.84 |
| | | **Ours (TPP) + QCFS** | ANN-SNN | 16 | 72.03 |
| | | **Ours (TPP)+ SNNC w/o Cali.** | ANN-SNN | 32 | 74.07 |

[*] The average number of timesteps during inference on the test dataset.

## 4.3 SPIKE ACTIVITY

The event driven nature of various neuromorphic chips implies that the energy consumption is directly proportional to the spiking activity, i.e., the number of spikes produced throughout the network: the energy is consumed in the presence of spikes. To this end, we tested our proposed method (TPP) for the spike activity and compared with the baselines. For a given model, we counted the average number of spikes produced after each layer, per sample, for both the baseline and our method. Figure 5 shows the example of RTS and RTS + TPP. Both the baseline and our method exhibit similar spike counts. In particular, our method constantly outperforms the baselines, and possibly in doing so it needs longer average latency per sample $(T + c)$. However, the energy consumed is approximately the same as that for the baseline in time $T$. The complete tables are present in Appendix E.4, where we provide more detailed picture of spike activities.

## 5 CONCLUSIONS AND FUTURE WORK

This work identified the phenomenon of "temporal misinformation" in ANN-SNN conversion, where random spike rearrangement enhances performance. We introduced two-phase probabilistic (TPP) spiking neurons, designed to intrinsically perform the effect of spike permutations. We show biological plausibility of such neurons as well as the hardware friendlines of the underlying mechanisms. We demonstrate their effectiveness through exhaustive experiments on large scale datasets, showing their competing performance compared to SOTA ANN-SNN conversion and direct training methods.

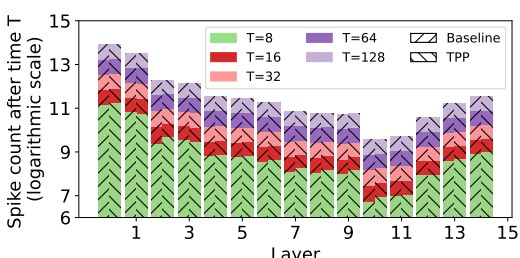

Figure 4: Spike counts of VGG-16 on CIFAR-100 of RTS baseline compared with RTS+TPP. Note: The bar height from bottom indicates the spike counts after each timestep T (see Appendix E.4)

In the future work, we aim to study the effect of permutations and probabilistic spiking in combination with directly trained SNN models.

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

## A    CONVERSION STEPS

**Copying ANN architecture and weights.**    ANN-SNN conversion process starts with a pre-trained ANN model, whose weights (and biases) will be copied to an SNN model following the same architecture. In this process, one considers ANN models whose non-activation layers become linear during the inference. In particular, these include fully connected, convolutional, batch normalization and average pooling layers.

**Approximating ANN activation functions.**    The second step of the process considers the activation layers and their activation functions in ANN. Here, the idea is to initialize the spiking neurons in the corresponding SNN layer in such a way that their average spiking rate approximates the values of the corresponding activation functions. For the ReLU (or ReLU -like such as quantized or thresholded ReLU ) activations, this process is rather well understood. The spiking neuron threshold is usually set to correspond to the maximum activation ANN channel or layerwise, or to be some percentile of it. If we denote by $f$ the ANN actiavtion, then ideally, after setting the thresholds, one would like to have

$$f(\mathbf{v}[T]) \approx \frac{\theta}{T} \cdot \sum_{t=1}^{T} \mathbf{s}[t]. \tag{8}$$

If we recall the equations for the IF neuron (equations equation 1 in the article)

$$\mathbf{v}^{(l)}[t] = \mathbf{v}^{(l)}[t-1] + \mathbf{W}^{(l)}\theta^{(l-1)} \cdot \mathbf{s}^{(l-1)}[t] - \theta^{(l)} \cdot \mathbf{s}[t-1], \tag{9}$$

$$\mathbf{s}^{(l)}[t] = H(\mathbf{v}^{(l)}[t] - \theta^{(l)}), \tag{10}$$

we see that the value with which we are comparing the membrane potential (threshold) is the same as the value with which we are scaling the output spikes. In particular, as soon as our membrane potential has reached $\theta$, it will produce the value $\theta$. This can be loosely described as, whatever the input is, the output will be approximately that value (or zero, if the input is negative), which is exactly what ReLU does.

**Absorbing thresholds.**    Finally, we notice that, once we produce a spike $\mathbf{s}^{(l)}[t]$, the value $\theta^{(l)} \cdot \mathbf{s}^{(l)}[t]$ will be sent to the next layer, and will further be weighted with weights $W^{(l+1)}$ and the bias $b^{(l+1)}$ will be applied. As we want SNNs to operate only using ones and zeros (to avoid multiplication due to energy efficiency), the values $\theta^{(l)}$ will be absorbed into $W^{(l+1)}$, i.e. $W^{(l+1)} \leftarrow \theta^{(l)}W^{(l+1)}$.

## B    PROOF OF THE THEORETICAL RESULTS

We prove the main theorems from the article, which we restate here.

**Theorem 1.** *Suppose that for some $0 < t < T$, we have $t \cdot \theta^{(l)} \leq \mathbf{v}^{(l)}[0] < (t+1) \cdot \theta^{(l)}$, and we are in the setting of equation 7. Then, the probability that the neuron will spike more than $t+1$ times, or less than $t$ times is zero. Moreover, the probability of having a spike at any given time step $t = 1, \ldots, T$ is non-zero.*

*Proof.* Notice that whenever there is a spike, the membrane potential decreases by $\theta^{(l)}$. In particular, after at most $t+1$ spikes, by the condition in the Theorem, the membrane potential will be negative. Hence, probability of having a spike will be 0. On the other side, if for $T - t$ time steps, we did not have a spike, this would mean that the bias $x$ of the Bernoulli variable $B(x)$ is larger than 1, which consequently will yield a spike with probability 1. Furthermore, after spiking, the bias remains bigger than 0. This means that we will have $t$ spikes with probability 1. The other cases are done in a similar way. The rest of the claim is easy.    $\square$

**Theorem 2.** *Suppose we have $N$ spiking neurons that produced spike trains $s_i[1], s_i[2], \ldots, s_i[T]$, $i = 1, \ldots, N$. Furthermore, suppose that these spike trains are modulated with weights $w_1, \ldots, w_N$, and as such give input to a neuron (say from the following layer) in the form $x[t] = \sum w_i s_i[t]$, for $t = 1, \ldots, T$. For a given permutation $\pi = (\pi_1, \ldots, \pi_N)$, let $\pi s_i$ denote the permutation of the spike train $s_i$. Then, for every $t_1, t_2 \in \{1, 2, \ldots, T\}$,*

$$E_\pi[\sum w_i \pi s_i[t_1]] = E_\pi[\sum w_i \pi s_i[t_2]].$$

*Proof.* It is enough to prove that for each $i = 1, \ldots, N$,

$$E_\pi[s_i[t_1]] = E_\pi[s_i[t_2]]. \tag{11}$$

Let $A(t_i)$ be the cardinality of the set of all the permutations that end up with a spike in step $t_i$, and note that the probability of having a spike at $t_i$ is then $\frac{A(t_i)}{T!}$. But, for each permutation that ends up with a spike at $t_i$, one can find a permutation that ends up with a spike at $t_2$ (by simply applying a cyclic permutation) and moreover this correspondence is bijective. In particular $A(t_i)$ is independent of $i$. The equation equation 11 and the statement follow. $\qquad\square$

## C    EXPERIMENTS DETAILS

### C.1    DATASETS

**CIFAR-10**: The CIFAR-10 dataset Krizhevsky et al. (2010) contains 60,000 color images of 32x32 pixels each, divided into 10 distinct classes (e.g., airplanes, cars, birds), with each class containing 6,000 images. The dataset is split into 50,000 training images and 10,000 test images.

**CIFAR-100**: The CIFAR-100 dataset Krizhevsky et al. (2010) consists of 60,000 color images of 32x32 pixels, distributed across 100 classes, with each class having 600 images. Similar to CIFAR-10, it is divided into 50,000 training images and 10,000 test images.

**ImageNet**: The ImageNet dataset Deng et al. (2009) comprises 1,281,167 images spanning 1,000 classes in the training set, with a validation set and a test set containing 50,000 and 100,000 images, respectively. Unlike the CIFAR datasets, ImageNet images vary in size and resolution. The validation set is frequently used as the test set in various applications.

### C.2    CONFIGURATION AND SETUPS

#### C.2.1    OURS + QCFS

**CIFAR**: We followed the original paper's training configurations to train ResNet-20 and VGG-16 on CIFAR-100. The Stochastic Gradient Descent (SGD) optimizer with a momentum of 0.9 was used. The initial learning rate was set to 0.02, with a weight decay of $5 \times 10^{-4}$. A cosine decay scheduler adjusted the learning rate over 300 training epochs. The quantization steps $L$ were set to 8 for ResNet-20 and 4 for VGG-16. All models were trained for 300 epochs.

**ImageNet**: We utilized checkpoints for ResNet-34 and VGG-16 from the original paper's GitHub repository. For ImageNet, $L$ was set to 8 and 16 for ResNet-34 and VGG-16, respectively.

#### C.2.2    OURS + RTS

**CIFAR**: We trained models using the recommended settings from the original paper.

**ImageNet**: We used pre-trained checkpoints for ResNet-34 and VGG-16 from the original paper's GitHub repository. Subsequently, all ReLU layers were replaced with spiking neuron layers.

For all datasets, we initialize TPP membrane potential to zero, while in the baselines we do as they propose.

#### C.2.3    OURS + SNNC W/O CALIBRATION

**CIFAR**: We adhered to the original paper's configurations to train ResNet-20 and VGG-16 on CIFAR-100. The SGD optimizer with a momentum of 0.9 was used. The initial learning rate was set to 0.01, with a weight decay of $5 \times 10^{-4}$ for models with batch normalization. A cosine decay scheduler adjusted the learning rate over 300 training epochs. All models were trained for 300 epochs with a batch size of 128.

**ImageNet**: We used pre-trained checkpoints for ResNet-34 and VGG-16 from the original paper's GitHub repository. Subsequently, all ReLU layers were replaced with our proposed spiking neuron layers.

# D  ALGORITHMS

The baseline SNN neuron forward function (Algorithm 1) initializes the membrane potential to zero and iteratively updates it by adding the layer output at each timestep. Spikes are generated when the membrane potential exceeds a defined threshold, $\theta$, and the potential is reset accordingly. This function captures the core dynamics of spiking neurons. The Shuffle Mode (Algorithm 2) is an extension of the baseline forward function. After generating the spikes across the simulation length, this mode shuffles the spike train.

The TPP Mode (Algorithm 3) introduces a probabilistic component to the spike generation process. Instead of a deterministic threshold-based spike generation, it uses a Bernoulli process where the probability of spiking is determined by the current membrane potential relative to the threshold adjusted for the remaining timesteps.

---

**Algorithm 1** SNN Neuron Forward Function and Additional Modes

**Require:** SNN Layer $\ell$; Input tensor $\mathbf{x}$; Threshold $\theta$; Simulation length $T$.
1: **function** BASELINESNN($\ell, \mathbf{x}, \theta, T$)
2:     $\mathbf{v} \leftarrow 0$ {Initialize membrane potential}
3:     **for** $t = 1$ **to** $T$ **do**
4:         $\mathbf{v} \leftarrow \mathbf{v} + \ell(\mathbf{x}(t))$
5:         $\mathbf{s} \leftarrow (\mathbf{v} \geq \theta) \times \theta$
6:         $\mathbf{v} \leftarrow \mathbf{v} - \mathbf{s}$
7:         Store $\mathbf{s}(t)$
8:     **end for**
9:     **return** $\mathbf{s}$
10: **end function**

---

**Algorithm 2** SNN Neuron Forward Function of Shuffle Mode

**Require:** SNN Layer $\ell$; Input tensor $\mathbf{x}$; Threshold $\theta$; Simulation length $T$.
1: **function** SHUFFLEMODE($\ell, \mathbf{x}, \theta, T$)
2:     $\mathbf{v} \leftarrow 0$ {Initialize membrane potential}
3:     **for** $t = 1$ **to** $T$ **do**
4:         $\mathbf{v} \leftarrow \mathbf{v} + \ell(\mathbf{x}(t))$
5:         $\mathbf{s} \leftarrow (\mathbf{v} \geq \theta) \times \theta$
6:         $\mathbf{v} \leftarrow \mathbf{v} - \mathbf{s}$
7:         Store $\mathbf{s}(t)$
8:     **end for**
9:     Shuffle the stored spikes $\mathbf{s}(1), \mathbf{s}(2), \ldots, \mathbf{s}(T)$
10:     **return** shuffled $\mathbf{s}$
11: **end function**

---

**Algorithm 3** SNN Neuron Forward Function of TPP Mode

**Require:** SNN Layer $\ell$; Input tensor $\mathbf{x}$; Threshold $\theta$; Simulation length $T$.
1: **function** TPPMODE($\ell, \mathbf{x}, \theta, T$)
2:     $\mathbf{v} \leftarrow \sum_{t=1}^{T} \mathbf{x}(t)$ {Initialize membrane potential with the sum of inputs}
3:     **for** $t = 1$ **to** $T$ **do**
4:         $\mathbf{p} \leftarrow \text{Clamp}(\mathbf{v}/(\theta \times (T - t + 1)), 0, 1)$
5:         $\mathbf{s} \leftarrow \text{Bernoulli}(\mathbf{p}) \times \theta$
6:         $\mathbf{v} \leftarrow \mathbf{v} - \mathbf{s}$
7:         Store $\mathbf{s}(t)$
8:     **end for**
9:     **return** $\mathbf{s}$
10: **end function**

---

# E  ADDITIONAL EXPERIMENTS

## E.1  SNNC

We show extra experiment results about the comparison among permutation method and two-phase probabilistic method. We validated ResNet-20 and VGG-16 on the CIFAR-10/100 dataset , and ResNet-34, VGG-16 and RegNetX-4GF on ImageNet with batch and channel-wise normalization enabled. Using a batch size of 128, the experiment was run five times with different random seeds to ensure reliable and reproducible results.

Table 4: Comparison between our proposed methods and ANN-SNN conversion SNNC method on **CIFAR-10**. The average accuracy and standard deviation of the TPP method are reported over 5 experiments.

| Architecture | Method | ANN | T=1 | T=2 | T=4 | T=8 | T=16 | T=32 | T=64 |
|---|---|---|---|---|---|---|---|---|---|
| ResNet-20 | SNNC-AP Li et al. (2021) | 96.95 | 51.20 | 66.07 | 83.60 | 92.79 | 95.62 | 96.58 | 96.85 |
| | **Ours (Permute)** | 96.95 | 34.05 | 61.46 | 90.54 | 95.05 | 96.12 | 96.62 | 96.77 |
| | **Ours (TPP)** | 96.95 | 10.05 (0.02) | 17.30 (0.52) | 79.19 (0.67) | 93.72 (0.05) | 95.87 (0.09) | 96.67 (0.04) | 96.80 (0.01) |
| VGG-16 | SNNC-AP Li et al. (2021) | 95.69 | 60.72 | 75.82 | 82.18 | 91.93 | 93.27 | 94.97 | 95.40 |
| | **Ours (Permute)** | 95.69 | 38.01 | 64.40 | 84.65 | 92.24 | 92.80 | 93.33 | 94.10 |
| | **Ours (TPP)** | 95.69 | 11.46 (0.35) | 32.24 (1.40) | 86.85 (0.42) | 94.34 (0.12) | 94.86 (0.06) | 95.48 (0.03) | 95.60 (0.04) |

Table 5: Comparison between our proposed methods and ANN-SNN conversion SNNC method on **CIFAR-100**. The average accuracy and standard deviation of the TPP method are reported over 5 experiments.

| Architecture | Method | ANN | T=1 | T=2 | T=4 | T=8 | T=16 | T=32 | T=64 |
|---|---|---|---|---|---|---|---|---|---|
| ResNet-20 | SNNC-AP Li et al. (2021) | 81.89 | 17.91 | 34.08 | 54.78 | 72.28 | 78.57 | 81.20 | 81.95 |
| | **Ours (Permute)** | 81.89 | 5.64 | 19.54 | 52.46 | 75.21 | 79.76 | 81.12 | 81.52 |
| | **Ours (TPP)** | 81.89 | 1.94 (0.11) | 5.15 (0.44) | 39.67 (0.99) | 71.05 (0.68) | 78.97 (0.24) | 81.06 (0.05) | 81.61 (0.08) |
| VGG-16 | SNNC-AP Li et al. (2021) | 77.87 | 28.64 | 34.87 | 50.95 | 64.30 | 71.93 | 75.39 | 77.05 |
| | **Ours (Permute)** | 77.87 | 12.50 | 34.98 | 60.81 | 69.42 | 72.78 | 73.50 | 75.14 |
| | **Ours (TPP)** | 77.87 | 2.05 (0.27) | 15.90 (0.71) | 59.23 (0.65) | 73.16 (0.17) | 76.05 (0.26) | 77.16 (0.09) | 77.56 (0.13) |

Table 6: Comparison between our proposed methods and ANN-SNN conversion SNNC method on ImageNet. The average accuracy and standard deviation of the TPP method are reported over 5 experiments.

| Architecture | Method | ANN | T=4 | T=8 | T=16 | T=32 | T=64 | T=128 |
|---|---|---|---|---|---|---|---|---|
| ResNet-34 | SNNC-AP Li et al. (2021) | 75.65 | – | – | – | 64.54 | 71.12 | 73.45 |
| | **Ours (Permute)** | 75.65 | 10.51 | 57.57 | 70.94 | 74.00 | 75.06 | 75.47 |
| | **Ours (TPP)** | 75.65 | 2.69 (0.03) | 49.24 (0.23) | 69.97 (0.10) | 74.07 (0.06) | 75.23 (0.03) | 75.51 (0.05) |
| VGG-16 | SNNC-AP Li et al. (2021) | 75.37 | – | – | – | 63.64 | 70.69 | 73.32 |
| | **Ours (Permute)** | 75.37 | 38.61 | 67.29 | 73.35 | 74.34 | 74.82 | 75.11 |
| | **Ours (TPP)** | 75.37 | 54.14 (0.59) | 69.75 (0.27) | 73.44 (0.02) | 74.72 (0.06) | 75.14 (0.02) | 75.25 (0.03) |
| RegNetX-4GF | SNNC-AP Li et al. (2021) | 80.02 | – | – | – | 55.70 | 70.96 | 75.78 |
| | **Ours (Permute)** | 78.45 | – | – | 43.45 | 68.12 | 75.63 | 77.63 |
| | **Ours (TPP)** | 78.45 | – | – | 22.71 (2.98) | 66.51 (0.44) | 75.54 (0.07) | 77.83 (0.04) |

## E.2 RTS

Table 7: Comparison between our proposed methods and ANN-SNN conversion RTS method on CIFAR-10/100 and ImageNet. The average accuracy and standard deviation of the TPP method are reported over 5 experiments.

| Dataset | Architecture | Method | ANN | T=4 | T=8 | T=16 | T=32 | T=64 | T=128 |
|---|---|---|---|---|---|---|---|---|---|
| CIFAR-10 | VGG-16 | RTS*Deng & Gu (2021a) | 94.99 | 88.64 | 91.67 | 93.64 | 94.50 | 94.76 | 94.91 |
| | | **Ours (Permute)** | 94.99 | 91.22 | 93.70 | 94.50 | 94.86 | 94.88 | 94.97 |
| | | **Ours (TPP)** | 94.99 | 91.49 (0.21) | 94.11 (0.09) | 94.72 (0.08) | 94.84 (0.06) | 94.91 (0.02) | 94.98 (0.02) |
| | ResNet-20 | RTS*Deng & Gu (2021a) | 91.07 | 27.08 | 40.88 | 65.13 | 84.75 | 90.12 | 90.76 |
| | | **Ours (Permute)** | 91.07 | 68.18 | 86.57 | 90.20 | 90.81 | 91.04 | 90.99 |
| | | **Ours (TPP)** | 91.07 | 72.87 (0.22) | 88.27 (0.14) | 90.44 (0.08) | 90.86 (0.14) | 90.94 (0.04) | 91.01 (0.03) |
| CIFAR-100 | VGG-16 | RTS*Deng & Gu (2021a) | 76.13 | 23.76 | 43.81 | 56.23 | 67.61 | 73.45 | 75.23 |
| | | **Ours (Permute)** | 76.13 | 35.31 | 62.84 | 71.20 | 74.34 | 75.53 | 75.92 |
| | | **Ours (TPP) + RTS** | 76.13 | 37.88 (0.35) | 65.81 (0.27) | 73.05 (0.12) | 75.17 (0.17) | 75.64 (0.12) | 75.90 (0.08) |
| ImageNet | VGG-16 | RTS Deng & Gu (2021a) | 72.16 | – | – | 55.80 | 67.73 | 70.97 | 71.89 |
| | | **Ours (Permute)** | 72.16 | 33.77 | 58.31 | 67.80 | 70.89 | 71.65 | 71.95 |
| | | **Ours (TPP)** | 72.16 | 30.50 (1.19) | 56.69(0.67) | 67.34 (0.25) | 70.63 (0.11) | 71.75 (0.05) | 72.05 (0.03) |

## E.3 QCFS

Table 8: Comparison between our proposed methods and ANN-SNN conversion QCFS method on CIFAR-10/100 and ImageNet. The average accuracy and standard deviation of the TPP method are reported over 5 experiments.

| Dataset | Architecture | Method | ANN | T=4 | T=8 | T=16 | T=32 | T=64 |
|---|---|---|---|---|---|---|---|---|
| CIFAR-10 | VGG-16 | QCFS*Bu et al. (2022c) | 95.76 | 94.33 | 95.21 | 95.65 | 95.87 | 95.99 |
| | | **Ours (Permute)** | 95.76 | 95.15 | 95.58 | 95.83 | 95.95 | 95.97 |
| | | **Ours (TPP)** | 95.76 | 95.28(0.09) | 95.84(0.1) | 95.95(0.05) | 95.98(0.06) | 95.97 (0.03) |
| | ResNet-20 | QCFS Bu et al. (2022c) | 92.43 | 79.45 | 88.56 | 91.94 | 92.79 | 92.82 |
| | | **Ours (Permute)** | 92.43 | 84.85 | 91.24 | 92.67 | 92.82 | 92.85 |
| | | **Ours (TPP)** | 92.43 | 86.24(0.18) | 92.08(0.11) | 92.70(0.1) | 92.78(0.04) | 92.68(0.06) |
| CIFAR-100 | VGG-16 | QCFS*Bu et al. (2022c) | 76.3 | 69.29 | 73.89 | 75.98 | 76.52 | 76.54 |
| | | **Ours (Permute)** | 76.3 | 74.28 | 75.97 | 76.54 | 76.60 | 76.64 |
| | | **Ours (TPP)** | 76.3 | 74.0(0.15) | 76.06(0.08) | 76.37(0.1) | 76.55(0.09) | 76.51(0.07) |
| | ResNet-20 | QCFS Bu et al. (2022c) | 67.0 | 27.44 | 49.35 | 63.12 | 66.84 | 67.77 |
| | | **Ours (Permute)** | 67.0 | 45.33 | 62.81 | 66.93 | 67.85 | 67.96 |
| | | **Ours (TPP)** | 67.0 | 47.0(0.2) | 64.66(0.25) | 67.28(0.12) | 67.61(0.1) | 67.77(0.06) |
| ImageNet | VGG-16 | QCFS Bu et al. (2022c) | 74.29 | – | – | 50.97 | 68.47 | 72.85 |
| | | **Ours (Permute)** | 73.89 | 55.54 | 71.12 | 73.65 | 74.28 | 74.28 |
| | | **Ours (TPP)** | 74.22 | 68.39 (0.08) | 72.99 (0.05) | 73.98 (0.07) | 74.23 (0.03) | 74.29 (0.00) |

## E.4 SPIKING ACTIVITY

The percentage difference between the baseline and our method in TPP mode is calculated as follows: Percentage Difference $= \frac{\text{Ours} - \text{Baseline}}{\text{Baseline}} \times 100$.

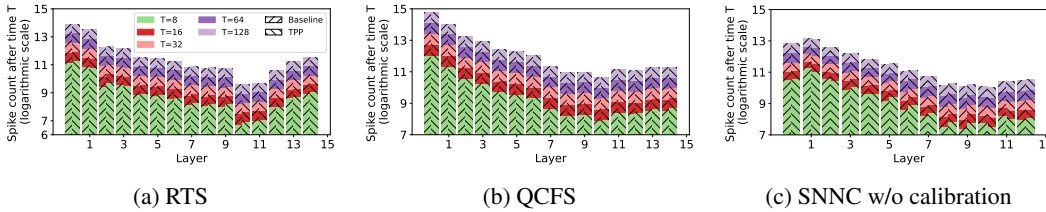

(a) RTS  (b) QCFS  (c) SNNC w/o calibration

Figure 5: Spike counts of VGG-16 on CIFAR-100 after different timesteps (T). Note: The bar height from bottom indicates the spike counts after each timestep T, and the color of longer Ts is overlaid by shorter Ts.

Table 9: Comparison of firing counts percentage difference between the baseline and our proposed TPP method for VGG-16 on CIFAR-100 using QCFS.

| Layer | T=4 | T=8 | T=16 | T=32 | T=64 | T=128 |
|---|---|---|---|---|---|---|
| 1 | 1.073 | 0.528 | 0.261 | 0.136 | 0.065 | 0.033 |
| 2 | 2.629 | 1.022 | 0.438 | 0.206 | 0.102 | 0.050 |
| 3 | 0.049 | 0.230 | 0.185 | 0.109 | 0.056 | 0.028 |
| 4 | -0.867 | -0.664 | -0.419 | -0.228 | -0.118 | -0.060 |
| 5 | 0.073 | 0.515 | 0.350 | 0.182 | 0.090 | 0.044 |
| 6 | 0.701 | 0.010 | -0.098 | -0.074 | -0.041 | -0.021 |
| 7 | -1.071 | -0.865 | -0.470 | -0.246 | -0.122 | -0.063 |
| 8 | 1.009 | 1.193 | 0.731 | 0.385 | 0.196 | 0.096 |
| 9 | 0.504 | 0.417 | 0.205 | 0.108 | 0.051 | 0.024 |
| 10 | -0.112 | 0.842 | 0.647 | 0.375 | 0.198 | 0.100 |
| 11 | 2.071 | 2.438 | 1.614 | 0.898 | 0.465 | 0.235 |
| 12 | 0.797 | 0.943 | 0.756 | 0.461 | 0.247 | 0.127 |
| 13 | 4.503 | 2.156 | 1.209 | 0.655 | 0.343 | 0.171 |
| 14 | 25.898 | 13.883 | 7.770 | 3.852 | 1.887 | 0.936 |
| 15 | 33.585 | 16.864 | 8.945 | 4.474 | 2.227 | 1.108 |

Table 10: Comparison of firing counts percentage difference between the baseline and our proposed TPP method for ResNet-34 on ImageNet using QCFS.

| Layer | T=4 | T=8 | T=16 | T=32 | T=64 | T=128 |
|---|---|---|---|---|---|---|
| 1 | 0.587 | 0.306 | 0.149 | 0.079 | 0.036 | 0.018 |
| 2 | -0.921 | -0.435 | -0.212 | -0.108 | -0.053 | -0.025 |
| 3 | 0.353 | 0.189 | 0.082 | 0.036 | 0.019 | 0.010 |
| 4 | -2.786 | -1.583 | -0.920 | -0.506 | -0.270 | -0.141 |
| 5 | 0.469 | 0.277 | -0.107 | -0.020 | -0.019 | -0.011 |
| 6 | -3.955 | -1.865 | -0.705 | -0.344 | -0.166 | -0.086 |
| 7 | -0.381 | 0.321 | -0.090 | -0.031 | -0.020 | -0.013 |
| 8 | 6.615 | 3.261 | 1.494 | 0.628 | 0.290 | 0.131 |
| 9 | -5.116 | -3.006 | -1.555 | -0.794 | -0.391 | -0.195 |
| 10 | -2.938 | 3.431 | 3.096 | 1.794 | 0.975 | 0.498 |
| 11 | 1.184 | 0.466 | 0.359 | 0.102 | 0.053 | 0.022 |
| 12 | -17.739 | -7.302 | -1.788 | -0.609 | -0.270 | -0.132 |
| 13 | 0.105 | -0.138 | -0.287 | -0.292 | -0.166 | -0.087 |
| 14 | -8.597 | -2.626 | 0.006 | 0.327 | 0.289 | 0.140 |
| 15 | -0.522 | -0.214 | -0.273 | -0.299 | -0.173 | -0.094 |
| 16 | -11.196 | -5.194 | -1.990 | -0.813 | -0.405 | -0.217 |
| 17 | -3.828 | -1.192 | -0.320 | -0.192 | -0.105 | -0.058 |
| 18 | -6.869 | -2.392 | -0.644 | 0.007 | -0.002 | 0.001 |
| 19 | 0.092 | -0.299 | -0.181 | -0.138 | -0.074 | -0.035 |
| 20 | -5.639 | -0.308 | 0.923 | 0.796 | 0.448 | 0.234 |
| 21 | 0.399 | -0.968 | -0.796 | -0.509 | -0.275 | -0.145 |
| 22 | -4.474 | 3.712 | 4.440 | 3.033 | 1.700 | 0.880 |
| 23 | 0.456 | -0.901 | -0.703 | -0.533 | -0.281 | -0.145 |
| 24 | -5.863 | 4.241 | 5.617 | 3.797 | 2.090 | 1.074 |
| 25 | 1.433 | -0.464 | -0.774 | -0.632 | -0.347 | -0.182 |
| 26 | -5.034 | 4.908 | 6.328 | 4.362 | 2.459 | 1.271 |
| 27 | 0.661 | -0.914 | -1.156 | -0.931 | -0.530 | -0.284 |
| 28 | -15.667 | 4.763 | 9.616 | 6.975 | 4.062 | 2.096 |
| 29 | -9.747 | 1.663 | 3.836 | 2.455 | 1.384 | 0.673 |
| 30 | -0.151 | 16.639 | 15.387 | 9.638 | 5.334 | 2.769 |
| 31 | -5.403 | 0.917 | 1.957 | 1.555 | 1.009 | 0.574 |
| 32 | 17.796 | 6.777 | 3.728 | 3.231 | 2.507 | 1.583 |
| 33 | -4.935 | -2.141 | 2.055 | 2.931 | 2.395 | 1.561 |

Table 11: Comparison of firing counts percentage difference between the baseline and our proposed TPP method for VGG-16 on ImageNet using QCFS.

| Layer | T=4 | T=8 | T=16 | T=32 | T=64 | T=128 |
|---|---|---|---|---|---|---|
| 1 | 5.487 | 2.776 | 1.444 | 0.712 | 0.363 | 0.179 |
| 2 | 0.418 | 0.173 | -0.005 | 0.007 | 0.007 | 0.006 |
| 3 | -2.375 | -0.883 | -0.351 | -0.128 | -0.062 | -0.031 |
| 4 | 6.170 | 2.181 | 0.627 | 0.121 | 0.024 | -0.002 |
| 5 | -3.338 | -0.318 | 0.327 | 0.306 | 0.173 | 0.097 |
| 6 | 7.036 | 2.769 | 0.993 | 0.385 | 0.173 | 0.078 |
| 7 | -5.722 | -3.482 | -1.661 | -0.800 | -0.400 | -0.200 |
| 8 | -6.155 | 0.310 | 1.411 | 0.955 | 0.507 | 0.269 |
| 9 | -0.718 | 1.172 | 0.725 | 0.337 | 0.162 | 0.081 |
| 10 | -12.833 | -9.060 | -4.882 | -2.359 | -1.145 | -0.564 |
| 11 | 12.966 | 11.241 | 7.718 | 4.443 | 2.344 | 1.188 |
| 12 | -11.194 | -14.874 | -12.032 | -7.889 | -4.437 | -2.395 |
| 13 | -37.388 | -30.782 | -20.701 | -12.296 | -6.527 | -3.377 |
| 14 | -23.619 | -12.312 | -3.929 | -0.233 | 0.585 | 0.382 |
| 15 | -10.988 | -18.476 | -13.953 | -7.904 | -4.091 | -2.015 |

## F    PERMUTATIONS AND STABILIZATION OF FIRING RATE

Table 12: Recorded accuracy after $t \leq T$ time steps, when the baseline model is "permuted" in latency $T$. Setting is VGG-16, CIFAR-100.

| Method | ANN | t=1 | t=2 | t=4 | t=8 | t=16 | t=32 |
|---|---|---|---|---|---|---|---|
| QCFS Bu et al. (2022c) | | 49.09 | 63.22 | 69.29 | 73.89 | 75.98 | 76.52 |
| **Ours (Permute)** | T=4 | 68.11 | 71.91 | 74.2 | | | |
| **Ours (Permute)** | T=8 | 71.76 | 74.11 | 75.53 | 75.86 | | |
| **Ours (Permute)** | T=16 | 72.75 | 74.27 | 75.63 | 76.0 | 76.39 | |
| **Ours ((Permute)** | T=32 | 73.15 | 75.23 | 75.74 | 76.27 | 76.59 | 76.52 |
| RTS Deng & Gu (2021b) | | 1.0 | 1.03 | 23.76 | 43.81 | 56.23 | 67.61 |
| **Ours (Permute)** | T=4 | 22.9 | 30.78 | 34.54 | | | |
| **Ours ((Permute)** | T=8 | 45.11 | 52.7 | 59.2 | 62.58 | | |
| **Ours ((Permute)** | T=16 | 54.58 | 64.37 | 68.6 | 70.8 | 71.79 | |
| **Ours (Permute)** | T=32 | 62.76 | 69.12 | 71.76 | 73.31 | 74.09 | 74.6 |

**Comments:**

1. In Table 12 we combine permutations with baseline models in fixed latency $T$. Afterwards, we record the accuracies of such "permuted" model for lower latencies $t$. We can notice a sharp increase in the accuracies compared to the baselines, and in particular, the variance in accuracies across $t$ is reduced.

2. **Baseline analysis:**

   (a) SNN models converted from a pretrained ANN aim to approximate the ANN activation values with firing rates. In particular, in lower time steps, the approximation is too coarse as the firing rate has only few possibilities to use to approximate the ANN (continuous) values. For example, in $T = 1$, the baselines are attempting to approximate ANN activations with binary values 0 and $\theta$.

   (b) Moreover, at each spiking layer, the spiking neurons at early time steps, use only the outputs of the previous spiking layer from the same, early, time steps. As this information is already too coarse, **the approximation error accumulates throughout the network**, finally yielding in models that are underperfoming in low latencies.

(c) With longer latencies, the model is using more spikes and is able to approximate the ANN values more accurately, and to correct the results from the first time steps.

3. **Effect of permutations:**

   (a) When performing permutations on spike trains after spiking layers in the baseline models, the input to the next spiking layer in lower time steps, **no longer depends only on the outputs of the previous layer in the same lower time steps, but it depends on the outputs in all time steps** $T$.

   (b) In particular, when spiking layer is producing spikes at time step $t = 1$, it does so "taking into account" (via permutation) outputs at all the time steps from the previous spiking layer.

   (c) As a way of example, consider two spiking neurons $N_1$ and $N_2$, where $N_2$ receives the weighted input from $N_1$. If a spiking neuron $N_1$ in one layer has produced spike train $s = [1, 0, 0, 0]$, in approximating ANN value of .25, then a spiking neuron $N_2$ at the first time step will use 1 as the approximation and will receive the input $W \cdot 1$ from neuron $N_1$. However, after a generic permutation of $s$, the probability of having zero at the first time step of output of neuron $N_1$ is $\frac{3}{4}$ (as oppose to having 1 with probability $\frac{1}{4}$), and at the first time step neuron $N2$ will most likely receive the input $W \cdot 0 = 0$ from neuron $N_1$, which is a rather better approximation for $W \cdot .25$ than $W$ itself.

   (d) This property of receiving input at lower $t$ but taking into account the previous layer spike outputs at all the time steps is not only exclusive to lower $t$. Indeed, at every time step $t \leq T$, the input at a spiking layer is formed by taking into account spiking train outputs from the previous layer at all the time steps, but having already accounted for for the observed input at the first $t < 1$ steps.

   (e) In general, the permutations overall increase the performance of the baselines because the spike trains are "uniformized" in accordance to their rate, and the accumulation error is reduced. If a layer $l$ has produced spike outputs that well approximate the $l$ layer in ANN, then, after a generic permutation, at each time step starting with the first, the next layer is receiving the most likely binary approximation of those rates.

   (f) This is nothing but Theorem 2 in visible action.

   (g) Besides Table 12, we provide further evidence on how permutation affect the baselines through the observed membrane potential in the following Appendix.

# G    MEMBRANE POTENTIAL DISTRIBUTION

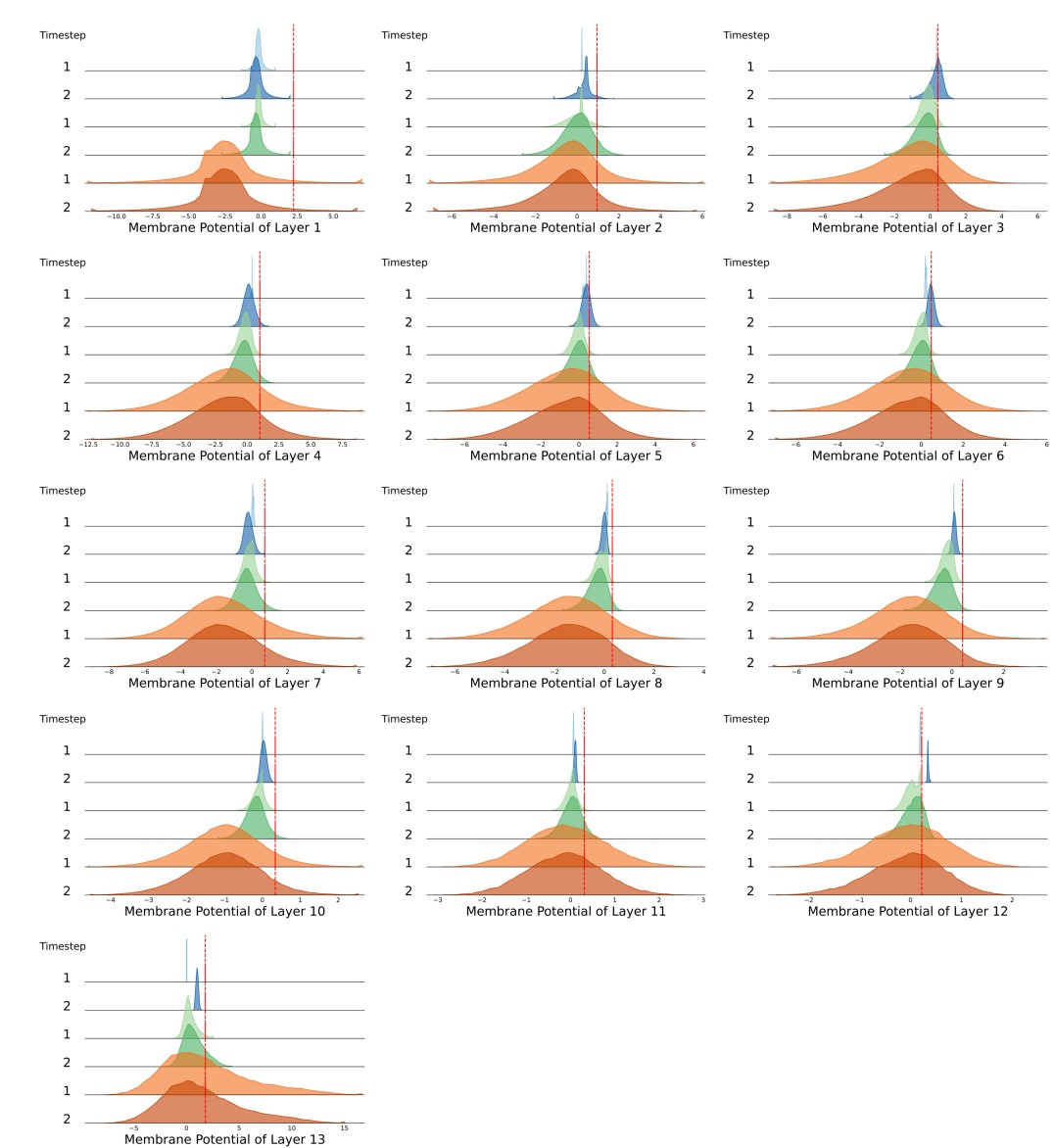

Figure 6: The membrane potential distributions of the first channel (randomly selected) across three modes (baseline, shuffle, and probabilistic) in VGG-16 on CIFAR-100. For comparison, the first two timesteps (t=1, t=2) from a total of eight timesteps (T=8) are selected for each mode. The baseline mode (blue) achieves an accuracy of 24.22%, while the shuffle mode (light green) improves accuracy to 70.54%, and the probabilistic mode (dark orange) further increases accuracy to 73.42%. The distributions are shown before firing, and the red dashed line indicates the threshold voltage (Vth) for the layer.

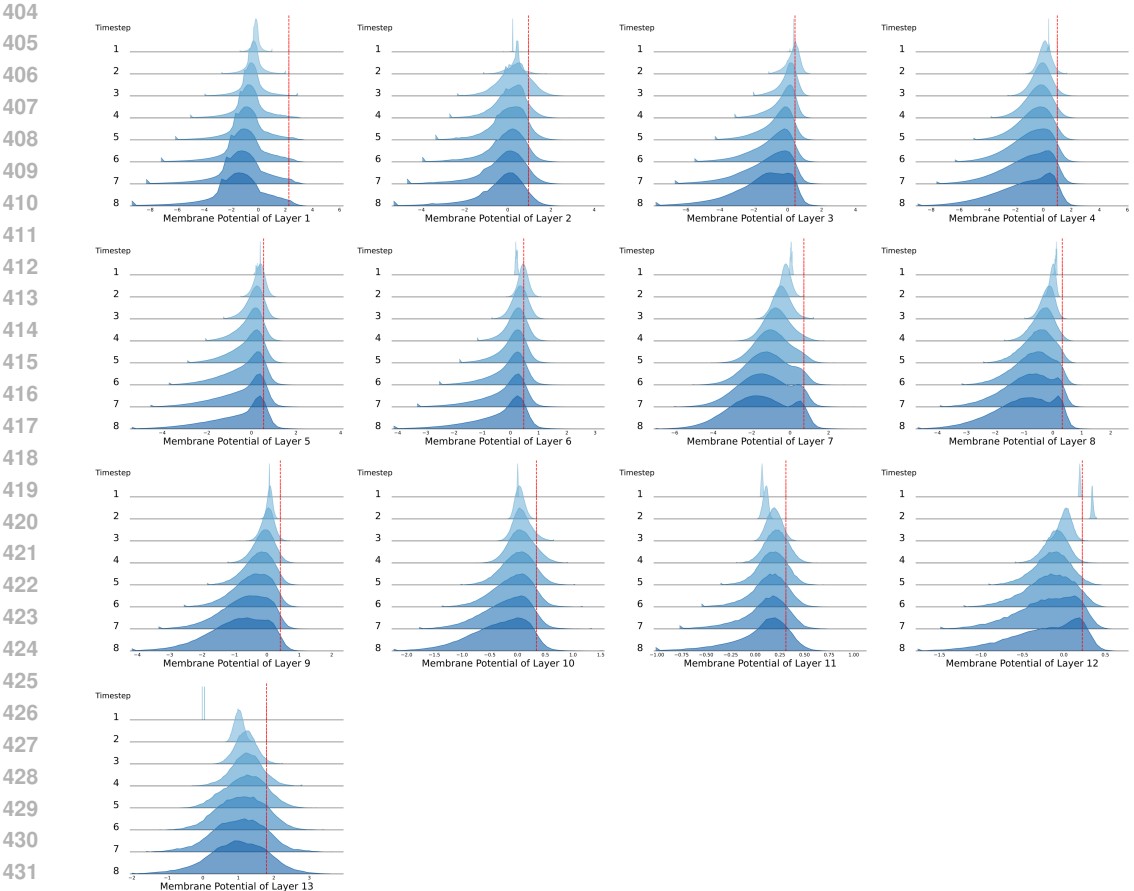

Figure 7: The membrane potential of the first channel (randomly selected) from layer 1 in SNNC baseline mode using VGG-16 on CIFAR-100 achieves an accuracy of 24.22% before firing.

The first two timesteps exhibit an abnormal distribution compared to those at t=4 to t=8. This discrepancy arises from the initially incorrect membrane potential before firing, which affects the firing rate and propagates errors layer by layer. A detailed quantifiable error analysis is provided in Appendix Section I. Furthermore, as shown in Figure 8, shuffling the membrane potential effectively alleviates this effect.

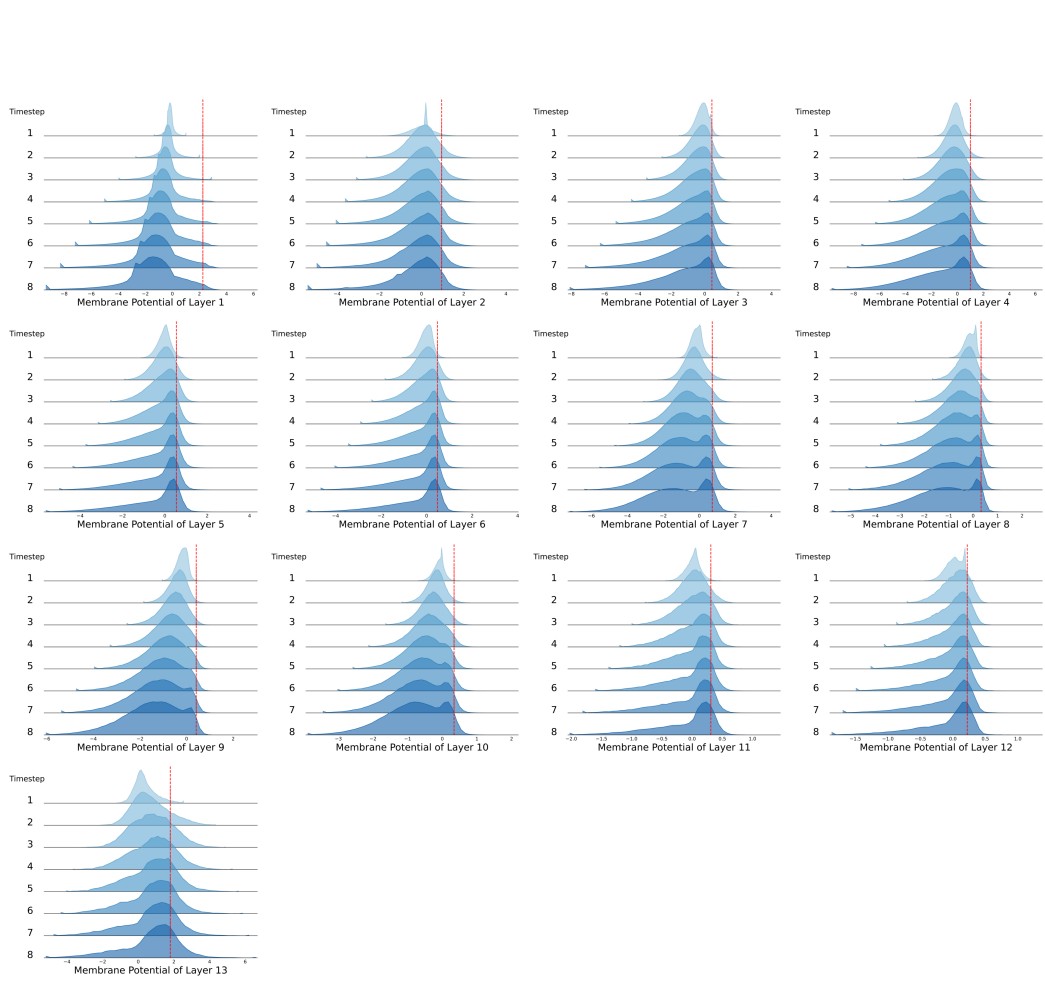

Figure 8: Membrane potential of the first channel (randomly selected) before firing in SNNC shuffle mode using VGG-16 on CIFAR-100. The achieved accuracy is 70.54%, indicating the impact of random spike rearrangement.

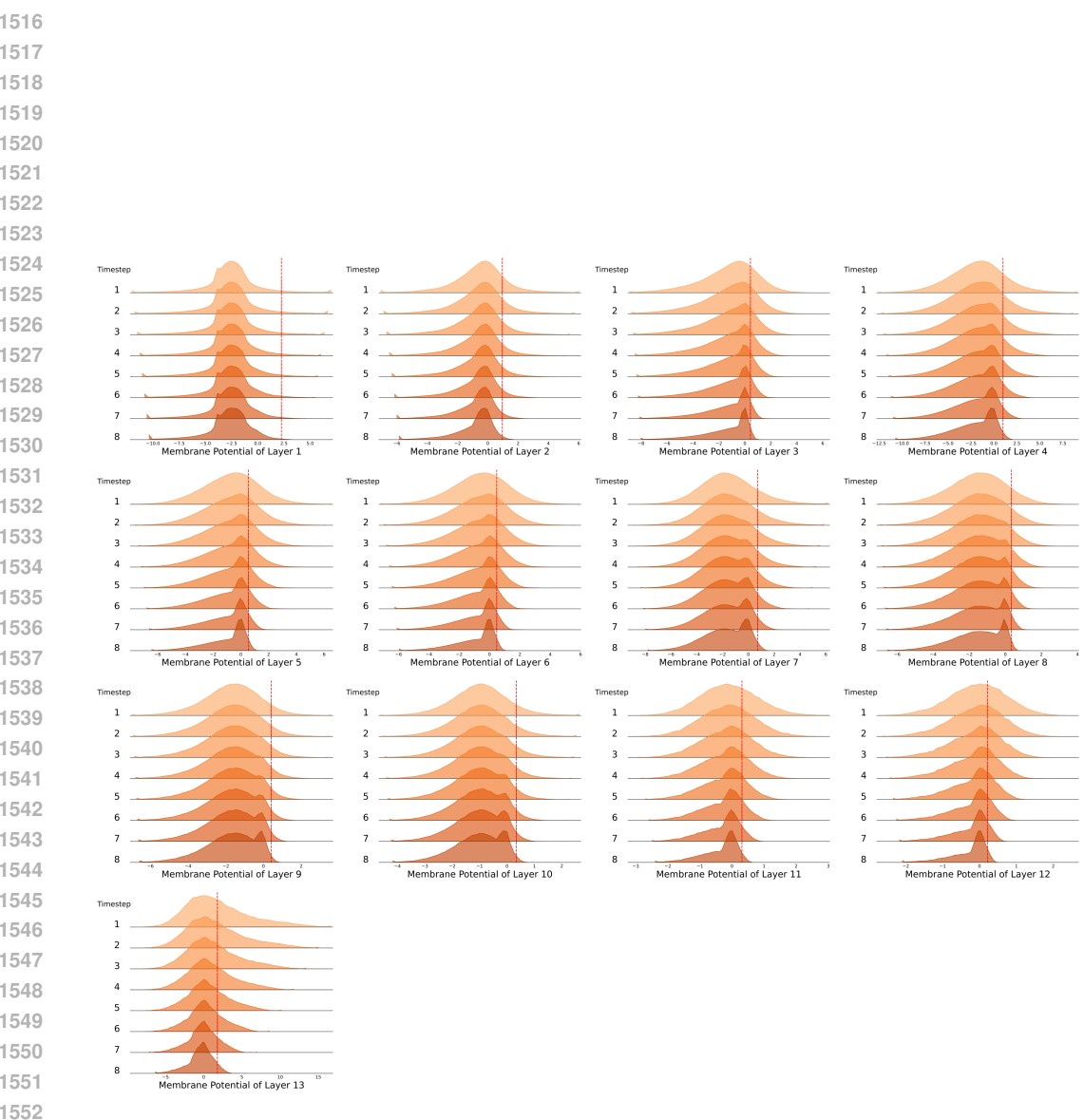

Figure 9: Membrane potential of the first channel (randomly selected) before firing in SNNC probabilistic mode using VGG-16 on CIFAR-100. The accuracy increases to 73.42%.

## H   THE EFFECT OF PERMUTATIONS ON PERFORMANCE: FURTHER EXPERIMENTS

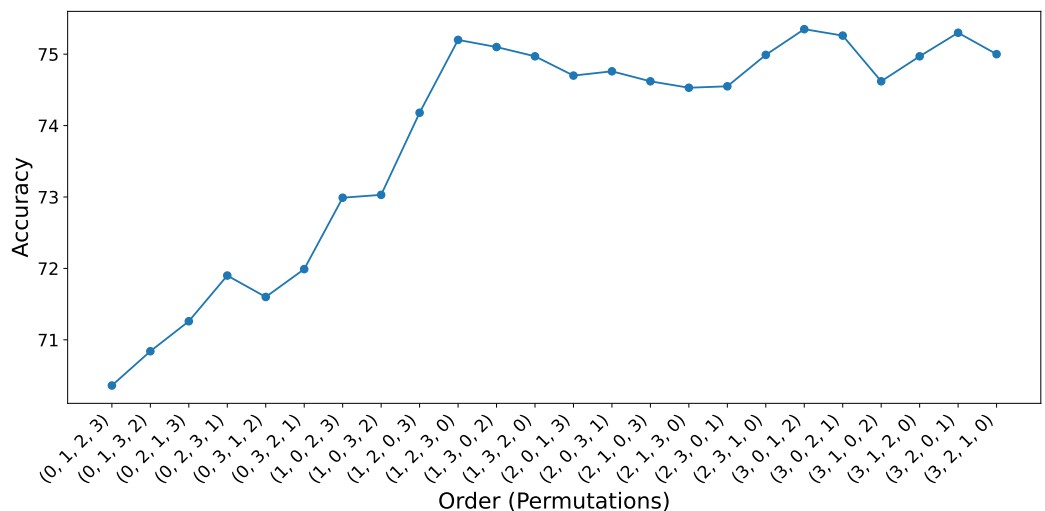

Figure 10: Accuracy comparison for all $T!$ permutations of input order over $T = 4$ time steps using QCFS with VGG-16 on CIFAR-100. Results of permuted orders outperform the original, non-permuted order $(0, 1, 2, 3)$. Baseline accuracy is $69.31\%$, The ANN accuracy is $76.21\%$.

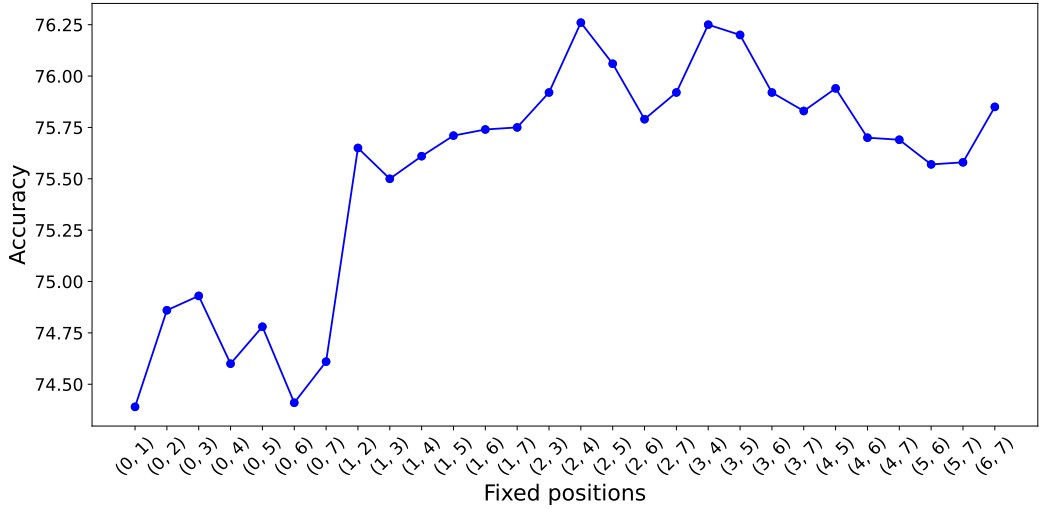

Figure 11: Accuracy comparison for permutations over 8 time steps, fixing given pairs of time steps. Setting is VGG-16, CIFAR-100. The baseline (QCFS) accuracy is $73.89\%$, ANN accuracy is $76.21\%$.

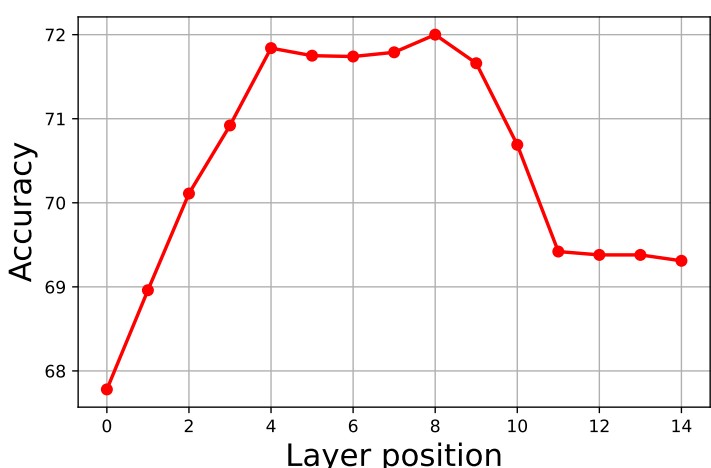

Figure 12: Accuracy of the model when a permutation is applied on a single layer using QCFS baseline. Setting is VGG-16, $T = 4$, CIFAR-100. Baseline accuracy is $69.31\%$, ANN accuracy is $76.31\%$

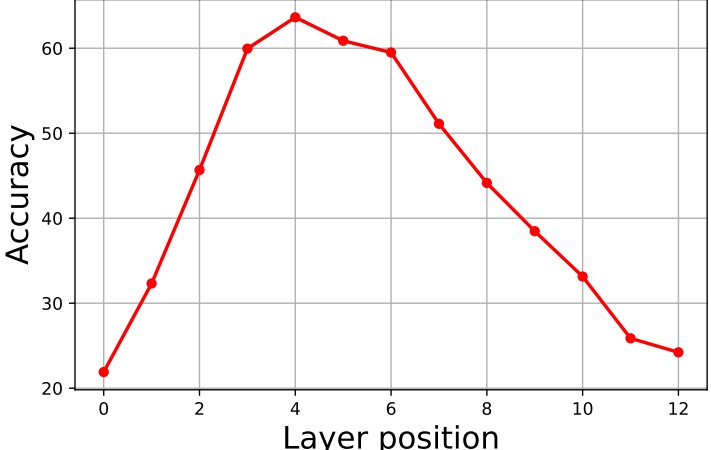

Figure 13: Accuracy of the model when a permutation is applied on a single layer using SNNC baseline. Setting is VGG-16, $T = 8$, CIFAR-100. Baseline accuracy without calibration is $24.22\%$, ANN accuracy is $77.87\%$

# I CONVERSION ERROR ANALYSIS

For this section, we use the terminology of Bu et al. (2022c) for the classification of conversion errors. We shortly recall three classes and we refer the reader to the original paper for more details:

1. **Clipping error**: When performing the ANN-SNN conversion, one uses some heuristics to set up the threshold, based on the corresponding distribution of the activation values. In particular, if $\mathcal{A}$ is the ANN activation, and $\theta$ is the set threshold for this particular layer, then the clipping error manifest itself in approximating $\mathcal{A}(\cdot)$ with $\min(\mathcal{A}(\cdot), \theta)$ (which is the maximum output of the spiking layer (before normalization)).

2. **Quantization error**: As the spiking neurons produce discrete spikes (values $0$ or $\theta$ (before normalization)), the quantization error manifest itself in using $\frac{\theta}{T} \cdot \max(0, \lfloor \frac{T}{\theta} \cdot x \rfloor)$ (which is tentative output of the spiking neuron) to approximate $\mathcal{A}(x)$.

3. **The unevenness error**: This error potentially occur due to the non-uniformity of the input to the spiking neurons. In particular, it can happen that the neurons receive streams of positive input during certain time period, while receiving stream of negative input during another period. Ideally, two streams should cancel each other parts of each other, but, due to their temporal mismatch, the neurons fire superfluous spikes, or they do not fire enough spikes as they theoretically should.

To study what is the main source of errors when performing ANN-SNN conversion with TPP neurons, we consider in detail the situation of a single layer of ANN neurons, and corresponding layer of SNN TPP neurons. For a function $f$ and constant $c$, we denote by $f_c$ the clipping of $f$ by $c$, that is $f_c(x) = \min(f(x), c)$. For example, $\mathrm{ReLU}_\theta(x) := \min(\mathrm{ReLU}(x), \theta) = \min(\max(0, x), \theta)$.

**Theorem 3.** *Let $X^{(l)}$ be the input of the ANN layer with* $\mathrm{ReLU}$ *activation and suppose that, during the accumulation phase, the corresponding SNN layer of TPP neurons accumulated $T \cdot X^{(l)}$ quantity of voltage.*

(a) *For every time step $t = 1, \ldots, T$, we have*

$$\frac{\theta}{t} \cdot \mathbb{E}\left[\sum_{i=1}^{t} s^{(l)}[i]\right] = \mathrm{ReLU}_\theta(X^{(l)}). \tag{12}$$

(b) *Suppose that for some $t = 1, \ldots, T$, the TPP layer produced $s^{(l)}[1], \ldots, s^{(l)}[t-1]$ vector spike trains for the first $t-1$ steps, and the residue voltage for neuron $i$ is higher than zero. Then,*

$$\frac{\theta}{t}\left(\mathbb{E}\left[s_i^{(l)}[t]\right] + \sum_{i=1}^{t-1} s_i^{(l)}[i]\right) = \mathrm{ReLU}_\theta(X_i^{(l)}). \tag{13}$$

(c) *If $s^{(l)}[1], \ldots, s^{(l)}[T]$ are the output vectors of spike trains of the TPP neurons during $T$ time steps, then*

$$\frac{\theta}{T}\sum_{i=1}^{t-1} s_j^{(l)}[i] = \begin{cases} \mathrm{ReLU}_\theta(X_j^{(l)}), & \text{if } \mathrm{ReLU}_\theta(X_j^{(l)}) \text{ is a multiple of } \frac{\theta}{T}, \\ \frac{\theta}{T} \cdot \lfloor \frac{T}{\theta}\mathrm{ReLU}_\theta(X_j^{(l)}) \rfloor \text{ or } \frac{\theta}{T} \cdot \lfloor \frac{T}{\theta}\mathrm{ReLU}_\theta(X_j^{(l)}) \rfloor + \frac{\theta}{T}, & \text{otherwise.} \end{cases} \tag{14}$$

(d) *Suppose that $\max X^{(l)} \leq \theta$ and that the same weights $W^{(l+1)}$ act on the outputs of layer $(l)$ of ANN and SNN as above, and let $X^{(l+1)}$ (resp. $T \cdot \tilde{X}^{(l+1)}$) be the inputs to the $(l+1)$th ANN layer (resp. the accumulated voltage for the $(l+1)$th SNN layer of TPP neurons), Then*

$$||X^{(l+1)} - \tilde{X}^{(l+1)}||_\infty \leq ||W^{(l+1)}||_\infty \cdot \frac{\theta}{T}. \tag{15}$$

**Comments:**

(a) We contrast this result with Theorem 2 of Bu et al. (2022c). Namely, there the authors show that if one uses half of the threshold as the initialization of the membrane potential,

the expectation of the conversion error (layerwise) is 0. However, the authors in Bu et al. (2022c) use the underlying assumption that the distribution of the ANN values layerwise is **uniform**, which in practice is not the case (see for example Bojkovic et al. (2024)). Our result $(a)$ above shows that **after every $t \leq T$ time steps**, our expected spiking rate aligns well with the clipping of the $\mathrm{ReLU}$ activation by the threshold, as it should, without any prior assumptions on the distribution of the ANN activation values.

(b) The point of result $(b)$ is that the activity of TPP neuron adapts to the observed output it already produced. In particular, as long as the neuron is still active and contains residue membrane potential, the expectation of its output at the next time step takes into account the previously produced spikes and will yield the ANN counterpart.

(c) (d) The results $(c)$ and $(d)$ show that during the accumulation phase, the TPP neuron approximate well the ANN neurons with ReLU activation. In particular, the only remaining source of errors in layerwise approximation is the clipping error due to the set threshold $\theta$, and the quantization error due to the discrete outputs of the spiking neurons. We also note in Equation equation 15 the two possibilities of the output in the second case ("otherwise").

