# OpenReview forum: "Temporal Misinformation and Conversion through  Probabilistic Spiking Neurons"
_ICLR.cc/2025/Conference — Submitted to ICLR 2025_

### Official Review · Reviewer_7dmg · 2024-10-30

**Soundness:** 2
**Presentation:** 1
**Contribution:** 2
**Rating:** 3
**Confidence:** 5

**Summary:**

This paper identified a new phenomenon in SNNs, termed “temporal misinformation”, and proposed a solution. Results are validated on CIFAR-10, and CIFAR-100, and ImageNet

**Strengths:**

The overall problem identification in the introduction section sounds interesting. However, the writing of the paper is very poor and it is very hard to follow the paper to understand the motivation, method, and detailed strengths of the method. Also, I cannot find detailed descriptions of Figure 1, which prevents me from understanding the main motivation of this work.

Basically, I lost at the very beginning phase when I read this paper, so have not been able to identify the strengths.

I encourage the author to make major changes to the clearness of the paper and writing before any potential resubmission, so readers can understand the paper better.

**Weaknesses:**

1. Line 11. "age of large neural network models", what "age of large neural network models" means? Is this a scientific term that deep learning researchers usually use? When did this age start? Which paper used the term "large neural network models" before?

2. I searched for the "Fig" but did not find any results that refer to Figure 1 explains Figure 1. Could you make it clearer to readers where the explanation in Figure 1 is located in your paper? It is NOT friendly for reviewers as you want to use Figure 1 to attract attention but then when a reviewer tries to enjoy the detailed explanation of this figure, you just hide explanations.

3. Line 158 "\textbf{ANN-SNN} conversion" and Line 188 "\textbf{Direct training} This method a". Why "conversion "is not capitalized in Line 158, but "This" is capitalized in Line 188? What does "ANN-SNN" mean? Is this a term you created? If not, please cite papers that used "ANN-SNN" before, or explain this new term clearly.

4. Line 25. "Code is available on GitHub.". What do you mean by "Code is available on GitHub."? There is not any Github link in the paper. Please do not mention something that does not exist in the abstract.



--------------------------------------------------------
Updates After Rebuttal

After multiple rounds of discussion with the authors, only Point 2 has been addressed. Point 3 remains unresolved and has not been corrected in the latest paper version (19 Nov 2024). The other two points have not been responded to by the authors.

I have provided my final comments to the authors, requesting a revision of the paper to solve Point 3. I will not respond to any further comments from the authors, as the authors ignored my previous comments and it has been the second time I EXPLICITLY asked the author to correct it in the paper. I strongly suggest author read my comments sincerely and revise the paper, instead of continuing to give pointless discussions.

Since the authors have addressed only one out of the four issues I raised, I have kept the score unchanged but increased the confidence level from 4 to 5.

------------------------
Second Update After Rebuttal:

This update focuses on the author's unprofessional behavior. I am extremely frustrated with how the author treats the reviewer and disregards the reviewer's comments.

I pointed out that Figure 1 in the paper is never referred to even once in the text. This is the first figure of the paper and should be clearly explained in detail. In the latest version of the paper (submitted on November 19, 2024), the author addressed one issue by correcting "Table 1" to "Figure 1" and responded that Figure 1 is described in the Introduction section and in Section 3.1. However, neither of these sections explains Figure 1 in detail, and Figure 1 is still not referred to in Section 3.1.

Additionally, I noted an issue in Line 158 regarding the term "ANN-SNN" I explained that "ANN-SNN" is not a term commonly used by researchers and that the first word of the sentence should be capitalized. The authors questioned the reviewer's expertise and professionalism, claiming that "ANN-SNN" is a widely used scientific term and suggested I look it up on Google Scholar. This response was unprofessional. I checked 30+ SNN papers on Google Scholar and confirmed that "ANN-SNN" is not used; instead, the technique is commonly referred to as "ANN-SNN Conversion." I informed the authors of this and requested that they revise the term explicitly in the paper and not let the efforts of the reviewer be wasted. Despite this, the authors refused to make the revision, arguing that the correct term is used 32 times elsewhere in the paper and dismissing my comments as "pointless discussions."

Even after providing evidence and explicitly asking for the revision, the authors have not made the changes. As of November 20, 2024, the term "\textbf{ANN-SNN}" has still not been revised to "\textbf{ANN-SNN Conversion}," and the first word of the sentence ("conversion") remains uncapitalized. The authors' repeated dismissal of my comments as "pointless" and their refusal to revise the paper is baffling and disrespectful to the review process. Their behavior is very unprofessional.

If the authors continue to treat the reviewer like this, I will escalate this matter to the Area Chair to address why the authors failed to revise the paper after the issues were clearly identified and why they treated the reviewer in such a dismissive manner.

**Questions:**

See Weakness for detailed questions.

---

> ### Author Response · Authors · 2024-11-19
>
> Thank you for pointing out some typos in our paper. We will correct these in the updated version, hopefully making the presentation more clear. The Figure 1 you mention is erroneously referred as Table 1 in the paper. It is first referred to in the Introduction, in the very first paragraph we start to describe our method (starting with line 094). It is further commented on in Section 3.1. Unfortunately, even after thorough reading of the paper after it is finished, these kind of mistakes still manage to go unnoticed.
>
> We do not doubt good intentions in your comments, but we do question their professionalism. Especially in repeated questions on the same topics. But, what perplexed us the most is your question about what ANN-SNN means. This is a standard terminology in the area of Spiking Neural Networks (meaning Conversion of Artificial Neural Networks to Spiking Neural Networks, sometimes also abbreviated as ANN-to-SNN or similarly). A search on Google Scholar would give dozens of papers with ANN-SNN in their title. This comment/question from your side, and the fact that you gave confidence 4 to your assessment of our paper, makes us doubtful.
>
> Of course, in case you have any constructive remarks and questions that pertain to the method we proposed, we will gladly try to assess them.

---

> ### Comment · Reviewer_7dmg · 2024-11-19
> **Please expain "ANN-SNN"**
>
> I am disappointed that the authors did not address all the questions I raised in my initial review. Additionally, I find it concerning that despite being informed about issues such as mis-references, improper capitalization, and the use of unprofessional terms, the authors questioned the expertise of the reviewer. This response is unwarranted and does not contribute to a constructive dialogue.
>
> Regarding the term "ANN-SNN," the authors appear to misunderstand why it is not widely accepted as a scientific term in the context of SNN literature. As per the author's suggestion, I conducted a search on Google Scholar and reviewed approximately 30 papers. None of these papers used "ANN-SNN" as a standard term for ANN-to-SNN conversion technology. Instead, terms like "ANN-to-SNN conversion" or "ANN-SNN conversion" are commonly used in previous research.
>
> I hope this clarification provides sufficient context for the authors to understand the concerns raised and reflect on the professionalism of their submission.
>
> I recommend that the authors review at least one SNN paper before responding to the question I raised regarding the use of the term "ANN-SNN. I am very surprised that the author just asked me to review other papers on Google Scholar to help the author correct the identified problem of the author`s own paper. It is ridiculous and very unprofessional.

---

> > ### Author Response · Authors · 2024-11-19
> >
> > Apart for thanking you for pointing to us the typos in the text, we do not see ground for constructive dialogue on this topic.
> >
> > We do not understand your objection to using the term ANN-SNN conversion. In weakness 3 you asked "What does "ANN-SNN" mean? Is this a term you created?" We simply pointed to the fact that ANN-SNN is a well-known and well-established abbreviation that many of the authors use. Our suggestion to perform a search on google scholar was just there to convince you of this fact.
> >
> > When you say "Instead, terms like "ANN-to-SNN conversion" or "ANN-SNN conversion" are commonly used in previous research" in your reply, this makes us think that your objection was because we used "ANN-SNN" rather than complete "ANN-SNN conversion" term? But then again, from 30+ time we used ANN-SNN in the text of the article, only once we did not follow it with conversion.
> >
> > Anyways, we are grateful for the insights about terminology, but we remain sceptic about your expertise in training SNNs with ANN-SNN (conversion).

---

> ### Comment · Reviewer_7dmg · 2024-11-19
> **Please correct "ANN-SNN" first**
>
> According to the author`s response, it seems that the author has understood why "ANN-SNN" is not a scientific term that is usually used. This is great, and I am happy that the author finally understands this. Please make sure to revise it correctly, as it is where a scientific term is introduced and explained in the paper. I read 30 papers to help the author solve this issue, I do not want my efforts to mean anything to the author, and this problem is still not revised.
>
> Reply to "Anyways, we are grateful for the insights about terminology, but we remain sceptic about your expertise in training SNNs with ANN-SNN (conversion)." Good, thanks for your replay and it is OK that you keep your "sceptic" (though I guess you mean skeptic )

---

> > ### Author Response · Authors · 2024-11-19
> >
> > Without any intention to engage in pointless discussions, we would like to say that your assertion “it seems that the author has understood why ‘ANN-SNN’ is not a scientific term” is baseless and misrepresenting of our position. Out of 33 times we used "ANN-SNN" in the text of our article, 32 times the term was followed by "conversion" and only one time by "setting".
> > Also, sceptic and skeptic are both valid spellings.

---

> > > ### Comment · Reviewer_7dmg · 2024-11-19
> > > **Please correct "ANN-SNN" first**
> > >
> > > I reviewed the latest version of the paper titled "Rebuttal Revision Edit by Authors" submitted on 19 Nov 2024. Despite several rounds of discussions and my explicit requirements to revise the paper, the authors have not yet addressed this specific issue nor updated a new paper version. This will be my final comment on this point and I will NOT reply to the author`s response anymore: PLEASE make sure you understand exactly what I mean and revise "ANN-SNN" in Line 158 as follows.
> > >
> > > Suggested revision:
> > >
> > > From:
> > > \textbf{ANN-SNN} conversion leverages pre-trained ANNs
> > >
> > > To:
> > > \textbf{ANN-SNN Conversion} ANN-SNN conversion leverages pre-trained ANNs.
> > >
> > > Or
> > > \textbf{ANN-to-SNN Conversion} ANN-to-SNN conversion leverages pre-trained ANNs.
> > >
> > > Thank you.

---

> > > > ### Author Response · Authors · 2024-11-20
> > > >
> > > > **Weakness 1.**  We changed the first sentence of the Abstract to "In the context of increasingly large neural network models and their associated high energy consumption, Spiking Neural Networks (SNNs) present a compelling alternative to Artificial Neural Networks (ANNs) due to their energy efficiency and closer alignment with biological neural principles". We hope this resolves the issue with "age of large neural network models" that you pointed.
> > > >
> > > > **Weakness 2.** We changed two references to Fig. 1 from "Table 1" to "Figure 1".
> > > >
> > > > **Weakness 3.** We changed \textbf{ANN-SNN} conversion to \textbf{ANN-SNN conversion} and corrected the capitalization typos.
> > > >
> > > > **Weakness 4.** We removed the sentence "Code is available on github." from the Abstract.
> > > >
> > > > You explicitly asked in Weakness 3.  "What does "ANN-SNN" mean? Is this a term you created? If not, please cite papers that used "ANN-SNN" before, or explain this new term clearly.", and you expect us to understand from this that you were actually suggesting changing "\textbf{ANN-SNN} conversion" to "\textbf{ANN-SNN conversion}"? Only in your last reply you made your suggestion clear, and only then we were able to identify this weakness.

---

> > > > > ### Comment · Reviewer_7dmg · 2024-11-20
> > > > >
> > > > > Please do not continue to complain about my reviewer comments.
> > > > >
> > > > > I asked you to explain what is "ANN-SNN" and make sure you follow the common terminology of SNN research, or provide an explanation of "ANN-SNN" if you insist on using this uncommon term (as combining two nouns does not make sense by itself). However, you asked me to check papers on Google Scholar and explain it to you. It is already very disrespectful as you are the author and I am not.
> > > > >
> > > > > Then I asked you to correct "ANN-SNN" after the problem was identified, but you refused. Thus, I give several suggested ways for revision and for the second time ask you to correct "ANN-SNN". Now you complain that I am not giving you an example of how to correct "ANN-SNN" at the beginning. Please note again you are the author who should make an effort to solve the questions reviewers raised. I am not the author of this paper who should read papers in Google Scholar for you and revise for you.
> > > > >
> > > > > I doubt if your reply represents the opinions of all authors. If not, please discuss it with other authors before replying to me. It is very unprofessional. I am doing volunteering reviewing and I am not a babysitter for you.

---

### Official Review · Reviewer_Ub8U · 2024-11-02

**Soundness:** 2
**Presentation:** 2
**Contribution:** 3
**Rating:** 3
**Confidence:** 5

**Summary:**

This paper analyzes the assumption in ANN-SNN conversion that information transfer relies solely on spike firing frequency. By permuting spike sequences, the paper observes the issue of “temporal misinformation” within the temporal domain. Further, it proposes an accumulation -> firing approach to avoid temporal misinformation, establishing an efficient ANN-SNN training method. The method is validated on datasets such as CIFAR and ImageNet, demonstrating advantages in model performance and other aspects.

**Strengths:**

The paper presents a compelling perspective: there exists a phenomenon of “temporal misinformation” in the ANN-SNN conversion method.

**Weaknesses:**

The novelty. The Ideas in this paper are quite similar to those in at least two other papers.

Please See the Question Part.

**Questions:**

1.Prior to this paper, many works have discussed the equivalence between spiking neurons with temporal information and neurons with quantized activation values. This includes papers in the ANN-SNN domain [A], as well as works in the direct training domain [B]. Could you highlight the key innovations in your paper?

2.I noticed that you cited paper [A], pointing out precision issues in its conversion process. In fact, your methods appear quite similar to theirs; could you provide a comparison between your approach and theirs?

3.Could your method be applied to direct training, as in [B]? Would you be able to discuss the similarities, differences, and advantages or disadvantages of your method when used in direct training compared to ANN-SNN conversion?

[A]Hu, Y., Zheng, Q., Jiang, X., & Pan, G. (2023). Fast-SNN: fast spiking neural network by converting quantized ANN. IEEE Transactions on Pattern Analysis and Machine Intelligence.
[B]Luo, Xinhao, et al. "Integer-Valued Training and Spike-Driven Inference Spiking Neural Network for High-performance and Energy-efficient Object Detection." arXiv preprint arXiv:2407.20708 (2024).

---

> ### Author Response · Authors · 2024-11-19
>
> **Weakness 1 and Q1:**
>
> We respectfully disagree with the claim that the ideas in this paper are similar to those in the two referenced works ([A] and [B]). While our work lies within the ANN-SNN conversion domain, the approaches and focuses are fundamentally different:
>
>  1. [A]: This work specifically addresses ANN-SNN conversion using quantized activations in ANN, relying on a 2-bit quantizer ( b = 2 ) and tailored quantization techniques. In contrast, our work has nothing to do with this method and it applies to general ReLU activated ANNs without **quantization**. We further emphasize that quantization of activations yields ANN models which are underperforming compared to those trained using ReLU activation, which further underscore the general applicability of our method. Our method is already ready to use for a pretrained ANN model (whether pretrained with ReLU or quantized ReLU activation), unlike the method in [A] which is only applicable for quantized models. Furthermore, even within the realm of ANN-SNN conversion, our proposed method has nothing to do with the baseline [A]. We are pointing to a common drawback of baseline ANN-SNN methods, which has to do with instable firing rates in early time steps, and propose a remedy for it in form of probabilistic spiking neuron.
>
>  2. [B]: This paper focuses on direct training of spiking neural networks (SNNs) with integer-valued spike-driven training, followed by ANN-SNN conversion framework, which lies outside the scope of our work (except for the conversion part). Our research remains fully within the ANN-SNN conversion paradigm, with a focus on bridging the performance gap between ANN (pretrained with ReLU or quantized ReLU activations) and SNN by optimizing spike dynamics during the conversion process.
>
> These distinctions demonstrate that our work provides unique contributions within the ANN-SNN domain that are not addressed by the referenced papers. The novelty of our work lies in studying the effect of permutations on spike trains in converted SNN models, and temporal processing strategies to improve the accuracy and energy efficiency of SNNs in the conversion framework, without relying on quantization or direct training methods.
>
>
> **Q2:**
>
> There are significant differences in methodology and focus:
>
> 1.	Quantization vs. ReLU: Paper [A] explicitly uses a 2-bit quantizer to facilitate conversion, whereas our work does not rely on any prior quantization in ANN models. We do combine our proposed approach to some of the baselines that use quantization, but this is just to show its applicability and not because of the necessity. This fundamental methodological difference highlights our focus on optimizing temporal dynamics rather than precision quantization. We once again emphasize that the quantized models tend to have weaker performance compared to those trained with ReLU activatoins, which consequently leads to lower performing SNN models.
>
> 2.	Signed neurons in [A]: Another fundamental difference is that [A] use signed neurons to address the accumulation error in the conversion. In other words, they use spiking neurons that are designed to produce negative spikes as well. Our neurons produce only positive spikes, and in a probabilistic manner which yields them biologically plausible as we discussed in the paper.
>
> We do point that the paper [A] relies on direct combination of two ideas: Quantization and Signed Neurons, both of which appeared elsewhere prior to their publication.
>
>
> **Q3:**
>
> Our method has not been tested in direct training, as its primary focus lies in the ANN-SNN conversion framework. Direct training (which in [B] is a prerequisite to perform ANN-SNN conversion) and ANN-SNN conversion represent fundamentally different paradigms with distinct challenges and goals.
>
> 1. Differences: Direct training methods, such as the one described in [B], optimize spiking activity during training, often relying on integer-valued updates or other specific mechanisms. In contrast, our method focuses on post-training optimization of spike order and temporal dynamics during conversion from ANN to SNN.
>
> 2. Advantages of Our Approach: Unlike direct training, our method does not require retraining the network or modifying the training pipeline, making it highly scalable and applicable to pre-trained ANN models. Furthermore, our approach achieves significant improvements in accuracy and energy efficiency with minimal additional computational cost, as demonstrated in Appendix E.4.
>
> In summary, while our work is not directly comparable to direct training methods like [B], it offers unique advantages and contributions within the ANN-SNN conversion domain specifically, and domain of training of SNN generally, emphasizing scalability and efficiency.

---

> > ### Comment · Reviewer_Ub8U · 2024-11-26
> >
> > The author states that their contribution is focus on ANN-SNN feild. But I think there is no differences between this method and the method used in SNN directly training. I will raise my score of contribution.

---

> > > ### Author Response · Authors · 2024-11-27
> > >
> > > Thank you for raising the contribution score and your time.
> > >
> > > But we are unclear about your comment: "I think there are no differences between this method and the method used in SNN directly training." Could you provide more details to help us better understand your perspective about "no differences"?
> > >
> > >
> > > The main contributions of our work are: 1) Systematic study of the **effect of permutations on spike trains and model performance**, conceptualized as **Temporal misinformation**. 2) We further propose a tailored **two-phase probabilistic (TPP) spiking neuron** to be used in the conversion and show its advantages through both theoretical and empirical analysis.
> > >
> > > We do not see how these are related to direct training.
> > >
> > > Just to make thing further clear, we make explicit the distinction between Direct Training and ANN-SNN Conversion
> > >
> > > Although both direct training and ANN-SNN conversion aim to develop efficient spiking neural networks (SNNs), they differ in their approaches and objectives.
> > >
> > > ANN-SNN Conversion: This approach leverages pre-trained artificial neural networks (ANNs) and focuses on converting them into SNNs with minimal accuracy loss. The primary goal is to align ANN activations with SNN firing rates, ensuring that the converted SNN closely approximates the behavior of the original ANN. Methods such as weight normalization, dynamic threshold adjustment, and post-conversion calibration are employed to mitigate conversion errors such as clipping, quantization, and temporal mismatches.
> > >
> > > Direct Training: In contrast, direct training involves building and training SNNs from scratch using spike-based computations. This method relies on spatio-temporal backpropagation through time (BPTT) and surrogate gradient methods to handle the non-differentiable nature of spikes. Direct training optimizes both synaptic weights and dynamic neuronal parameters (e.g., firing thresholds, membrane leakage), allowing SNNs to fully exploit precise spike timing and temporal dynamics. However, it often faces challenges related to gradient instability and high computational costs.
> > >
> > > Both Direct training and ANN-SNN conversion are very rich research directions within the scope of SNNs and plenty of methods have been proposed on both sides. Our work is situated within the ANN-to-SNN conversion framework.

---

> ### Author Response · Authors · 2024-11-25
>
> As the deadline for the rebuttal is approaching, we wanted to kindly check if we have adequately addressed your concerns regarding our paper. If there are any outstanding issues or additional points that you feel need clarification, please do not hesitate to let us know.
>
> Authors

---

### Official Review · Reviewer_gkwM · 2024-11-04

**Soundness:** 2
**Presentation:** 2
**Contribution:** 3
**Rating:** 6
**Confidence:** 4

**Summary:**

This paper introduces the concept of "temporal misinformation" in the ANN-to-SNN conversion process. It employs two-phase probabilistic (TPP) spiking neurons as the neurons in the SNN. The converted SNN model improves performance by using probabilistic neurons that fires spikes randomly.

**Strengths:**

1. The method is relatively easy to understand.
2. The problem is relevant to the scope of ICLR.
3. The paper identifies an interesting phenomenon in the ANN-SNN conversion process, which could serve as a complementary insight.

**Weaknesses:**

1. This work primarily proposes a spiking neuron model rather than a novel conversion paradigm, which limits its originality, as similar work on probabilistic neurons has been done previously.
2. The authors’ definition of "temporal misinformation" is unclear. While the paper introduces the "temporal misinformation" phenomenon and experimentally demonstrates its impact on SNN performance, it lacks a sufficient theoretical basis to explain this phenomenon and does not provide quantifiable error analysis.
3. The authors use "permutation" to introduce two-phase probabilistic spiking neurons (TPP). Theorem 1 suggests that the spikes emitted by probabilistic spiking neurons achieve an optimal spike firing order; however, this proof lacks persuasiveness.

**Questions:**

1.  For T time steps, there are T! possible permutations. Does every permutation yield better results than the original, non-permuted approach?
2. Please clarify what the additional c steps in Tables 1 and 2 represent.
3. In Tables 4 and 5, when comparing with SNNC, data suggests that at short time steps such as T=4 and T=8, "Permute" seems more effective than TPP. I would like to see if a similar phenomenon occurs with QCFS. Please provide additional comparative experiments between "Permute" and TPP on QCFS across CIFAR-10 and CIFAR-100 datasets.
4.  Does the probabilistic neuron model introduce extra overhead for hardware implementation?

---

> ### Author Response · Authors · 2024-11-19
>
> **Weakness 1.** While we acknowledge that probabilistic spiking neurons have been explored in prior work within the general domain of Spiking Neural Networks, to the best of our knowledge, this is the first work to propose and rigorously analyze their usage within the ANN-SNN conversion paradigm. Beside this, we point to the common phenomenon in ANN-SNN methods, and exploit it for the tailored construction of our probabilistic neurons.
>
> **Weakness 2.** We hope that the common answer above clarifies the notion of temporal misinformation. As for the theoretical basis, we point to Theorem 1 of why the permutation increase the overall performance of the baseline models (uniformization of the input), but we also try to explain this in more details here.
>
> We claim that the reason behind temporal misinformation phenomenon is the instability of firing rates of baseline SNN models in early time steps.
>
> SNN models converted from a pretrained ANN aim to approximate the ANN activation values with firing rates. The approximation is given by $\mathbf{x}^{(\ell)} \approx \bar{\mathbf{s}}^{(\ell)} = \frac{1}{T} \sum_{t=0}^{T} \mathbf{s}^{(\ell)}(t)$. However, in lower time steps, the approximation is too coarse as we can only use few spikes in order to approximate the ANN (continuous) values. For example, in $T=1$, the baselines are attempting to approximate ANN activations with binary values $0$ and $\theta$.
>
> What is crucial is that, at each spiking layer, the spiking neurons at early time steps, use only the outputs of the previous spiking layer from the same, early, time steps. As this information is already too coarse, **the approximation error accumulates throughout the network**, finally yielding in models that are underperforming in low latencies.
>
> Finally, with longer latencies, the model is using more spikes and is able to approximate the ANN values more accurately, and to correct the results from the first time steps. (One can argue that at long enough latencies, all reasonable conversion methods work equally well.)
>
> How do permutations fix this common occurrence?
>
> When performing permutations on spike trains after a spiking layer in the baseline models, the input to the next spiking layer in lower time steps **no longer depends only on the outputs of the previous layer in the same, lower time steps, but it depends on the outputs in all time steps $T$**.
> In particular, when a spiking layer is producing spike at time step $t=1$, it does so "taking into account" (via permutation) outputs at all the time steps from the previous spiking layer.
>
> As a way of example, consider two connected spiking neurons $N_1$ and $N_2$, where $N_1$ is sending the weighted output to $N_2$. If a spiking neuron $N_1$ in one layer has produced spike train $s = [1,0,0,0]$, in approximating ANN value of $.25$, then a spiking neuron $N_2$ at the first time step will use 1 as the approximation and will receive the input $W\cdot 1$ from neuron $N_1$. However, after a generic permutation of $s$, the probability of having zero at the first time step of output of neuron $N_1$ is $\frac{3}{4}$ (as oppose to having 1 with probability $\frac{1}{4}$), and at the first time step neuron $N_2$ will most likely receive the input $W\cdot 0=0$ from neuron $N_1$, which is a rather better approximation for $W\cdot .25$ than $W \cdot 1$.
>
> This property of receiving input at lower $t$ but taking into account the spike outputs of the previous layer at all the time steps is not only exclusive to lower $t$. Indeed, at every time step $t\leq T$, the input at a spiking layer is formed by taking into account spiking train outputs from the previous layer at all the time steps, but having already accounted for for the observed input at the first $t-1$ steps.
>
> In general, the permutations overall increase the performance of the baselines because the spike trains are "uniformized" in accordance to their rate, and the accumulation error is reduced. If a layer $l$ has produced spike outputs that well approximate the $l$ layer in ANN, then, after a generic permutation, at each time step starting with the first, the next layer is receiving the most likely binary approximation of those rates.
>
> This is nothing but Theorem 2 of the paper in visible action.
>
> We provide several evidence for our claims (new appendices) which are as follows: Appendix F observes what is the accuracy of the model at $t=1$, when we applied permutations on spike trains of higher length $T$; Appendix G shows the membrane potential distribution before firing in the baselines and permuted and TPP models. Permuted and TPP models show higher variance, hence their enhancement of the model's ability to produce spikes in low latency. Appendix H, which shows the experiments when particular classes of permutations are applied on the baseline models. In particular, the permutation that anchor first time step show the least increment in performance.

---

> ### Author Response · Authors · 2024-11-19
>
> **Weakness 3.**
>
> We kindly point the Reviewer to the **newly added Appendix I** in our paper and Theorem 3, therein. We point that Theorem 3 ammeliorates the findings of Theorem 1 (which becomes a consequence of statement (c) in Theorem 3). We also point to the Comments following Theorem 3, where we contrast our results with some similar ones in the literature.
>
> **Q1:**
>
> In general, we do not think so. For example, in our early experiments we permuted each output spike train after every layer, separately. Later on, we apply the same permutation on every spike train that belongs to the same layer, and we did not see major differences in the performance of the two. However, if we stick to the first case, where each spike train is permuted independently, then it is not difficult to argue that carefully choosing permutations would degrade the performance of the model. For example, We can permute the spike trains which are modulated with positive weights in such a way that all the spikes are clustered in the beginning of the time frame, while we can permute the spike trains which are modulated with negative weights in such a way that all the spikes are clustered at the very end of the time frame. This way, the spike trains are completely misaligned and would not integrate together at the correct timings (in the result, every layer would produce superfluous spikes, in general).
>
> Having said this, we have conducted several additional experiments, detailed in Appendix H, to address the reviewer’s inquiry about the impact of spike order on performance.
>
> Our implementation is based on QCFS (T=4). The baseline accuracy of the ANN VGG-16 model is 76.21\%. For the SNN model, we evaluated all 24 possible permutations of the four timesteps to systematically analyze their effect on accuracy (on every spike train, in every layer, the same permutation has been applied in one run) . To our surprise, all the permutations increased the performance of the baseline model. Why did this happen is beyond our current understanding. However, we notice that the least performance increment comes from the permutations that are anchoring the first time step: configurations starting with the unshuffled 0-th timestep, such as (0, 1, 2, 3), (0, 1, 3, 2), and (0, 2, 1, 3), yielded the lowest accuracies among all permutations. This pattern suggests a significant sensitivity to the initial timestep, likely due to the role of early spikes in establishing the network’s dynamic state.
>
> Furthermore, we considered the same baseline for $T=8$ and we performed systematically permutations which were anchoring pairs of time steps $(i,j)$. This is in the Appendix Figure 11. Once again, the plot shows the the permutations that are fixing the first time step $(0,j)$ show the least performance increment.
>
> Finally, we also investigated the affect of permutations on a single layer of the baselines. Figure 12 and 13 show the effect of a permutation when it is applied only on a single layer in the baseline model. We notice that the permutation show the least effect when they are applied in the early layers (as the SNN model is still approximating the ANN well), and in the late layers (as it is already too late to fix the accumulated approximation error from the previous layers).
>
> **Q2:**
>
> Our TPP neuron model works in two phases. In the first phase it accumulates the incoming spike trains of length $T$. This operation of accumulation potentially can be done in less than $T$ steps (as the neuron is receiving them all at once) on specific hardware, and we used $c$ to denote this time.
>
> **Q3**
>
> We provide an additional experimental table in Appendix E.3, comparing our proposed methods with the ANN-SNN conversion QCFS method using VGG-16 and ResNet-20 on CIFAR-10, CIFAR-100, and ImageNet. The table includes a detailed comparison of accuracy and standard deviation, with results averaged over five experiments for the TPP method.
>
> Both the permutation and TPP methods demonstrate consistent and significant improvements in accuracy over the baseline QCFS approach and they are comparable between themselves. One visible case where TPP outperforms the permutations is on ImageNet dataset. These results highlight the effectiveness of our proposed methods in addressing the inherent limitations of traditional ANN-SNN conversion methods. Our methods achieve higher performance across diverse datasets and network architectures, underscoring their robustness and generalization.
>
> **Q4**
>
> The  energy cost for the random sampling which is crucial for the implementation of our TPP neurons is unfortunately not readily reported for neuromorphic chips we enlisted in the paper, but is usually integrated in the overall energy efficiency of the chip. However, any additional cost is compensated with the superior performance of our models, as they are achieving close to ANN performance two times or faster than the baselines, hence producing approximately twice as little spikes and SOPs.

---

> ### Author Response · Authors · 2024-11-25
>
> As the deadline for the rebuttal is approaching, we wanted to kindly check if we have adequately addressed your concerns regarding our paper. If there are any outstanding issues or additional points that you feel need clarification, please do not hesitate to let us know.
>
> Authors

---

> ### Comment · Reviewer_gkwM · 2024-11-25
>
> 1. Thank you for the effort put into this work. The results in Figures 10, 11, and 12 are quite interesting. The figures illustrate the accuracy under $T! $ different permutation scenarios. I am curious about that did the authors perform the $ T! $  tests on only a single layer? This is because I believe the search space would be $ O(N \cdot T!) $ , where $ N$ is the number of layers in the network. The results in Table 12 seem important, but I am still unclear about the meanings of $  T $  and $  t $  in the table. Could the authors provide further clarification?
>
> 2. I found the authors' response to weak3 very intriguing. Perhaps there exists some optimal permutation that could be identified through a certain search method. However, that might be a separate line of work.
>
> 3. If the authors can address my questions, I will raise my score by 1.

---

> ### Author Response · Authors · 2024-11-25
>
> Thank you for your comments and for the opportunity to clarify further some points.
>
> 1. We clarify the details for Figures 10, 11, 12 and 13 in order.
>
> Figure 10: We consider $T=4$ and there are  4!=24 possible permutations. Once a permutation is fixed (as can be observed on the $x$-axis), then it is consistently applied throughout all the layers  in the baseline model. Finally, the accuracy of the model is recorded.
>
> Figure 11: We consider $T=8$ and we group permutations according to the pair of time steps that they are leaving fixed. For example, if a pair $(i,j)$ is present on the $x$-axis, it means that we consider the permutations that are leaving time steps $i$ and $j$ fixed, and this is the only constraint on the permutation. Furthermore, at each layer, the choice of permutation (that is fixing $(i,j)$) is arbitrary, that is, permutations can vary from layer to layer as long as they are fixing $(i,j)$. Then, for each such fixed pair, we recorded the accuracy of the baseline model.
>
> Figures 12 and 13: We consider $T=8$ and two baseline methods (QCFS and SNNC, respectively). Then, for the respective baseline, we chose one layer (the position of the layer is on the $x$-axis) and we only permute the spike trains after that chosen layers (the permutation is arbitrary). The permutations are not applied on any other layers. Then, we observe the final accuracy of the model.
>
> We did not perform experiments where we apply all the possible $T!$ permutations on one single layer in the baseline models.
>
> Table 12: The goal of the experiments performed therein was to support our initial hypothesis for why permutations improve the performance of the baselines. As we wrote in the Comments following the table, when performing permutations on spike trains after spiking layers in the baseline models, the input to the next spiking layer in lower time steps, no longer depends only on the outputs of the previous layer in the same lower time steps, but it depends on the outputs in all time steps $T$.
>
> In particular, when spiking layer is producing spikes at time step $t=1$, it does so "taking into account" (via permutation) outputs at all the time steps from the previous spiking layer. To test that this is indeed the case, we performed experiments presented in Table 12. For a fixed latency $T$, and after applying permutations on spike trains in the baseline model, we recorded the accuracy of the ``permuted'' model at $t<T$. In other words, in the output layer, we do not take the whole spike trains of length $T$ but we cut them after $t$. Then we reported those accuracies. Because of permutations, the accuracies at all time steps $t$ are more stable and close to each other, as compared to the baseline models.
>
>
> 2. This indeed could be the case. We also find very interesting the results of Figures 12 and 13, where an effect of a single permutation at a particularly well chosen hidden layer could improve the performance of the model drastically. But, this, as you already hinted, could be a separate line of work.

---

> > ### Comment · Reviewer_gkwM · 2024-11-27
> >
> > Thank you for your efforts, I will increase the contribution score.

---

> ### Author Response · Authors · 2024-11-27
>
> Thank you for your review and for increasing the contribution score following our revisions. We appreciate the time you’ve taken to provide feedback on our work.
>
> We believe we have addressed the weaknesses and questions you initially raised, and we would like to kindly ask if there are any remaining concerns or areas in the paper that would need further improvement? Additionally, we were wondering if our past and potential new revisions might lead you to reconsider the overall evaluation score, as it currently places our work below the acceptance threshold.

---

> > ### Comment · Reviewer_gkwM · 2024-12-03
> >
> > Thank you for your efforts. Although my concerns have been partially addressed, I am willing to increase the rating score.

---

> > > ### Author Response · Authors · 2024-12-03
> > >
> > > Thank you very much for your time and effort in providing us review and comments on our paper, and for increasing the score.
> > >
> > > Authors

---

### Official Review · Reviewer_egA6 · 2024-11-04

**Soundness:** 3
**Presentation:** 1
**Contribution:** 3
**Rating:** 8
**Confidence:** 4

**Summary:**

This paper primarily conducts further research on the ANN-SNN conversion methods. In particular, the authors have discovered a new phenomenon termed "temporal error information" and proposed biologically plausible two-phase probabilistic (TPP) spiking neurons for ANN-SNN conversion. The experimental results demonstrate the advantages of this method.

I believe that the method proposed by the authors is effective; however, I have some questions that need clarification, mainly regarding the explanation and interpretation of the phenomenon. At this stage, my rating is "6: Marginally above acceptance threshold." If the authors could address and clarify these questions, I would be very willing to raise my score.

**Strengths:**

1. The method is simple and effective, supported by comprehensive experimental validation. The proposed approach achieves state-of-the-art (SOTA) accuracy on the CIFAR-10/100 and ImageNet datasets, advancing the further development of ANN-SNN conversion.

2. Unlike SNN research that tends to focus on computational efficiency, the two-phase mechanism and probabilistic spike discharge proposed in this work are both biologically plausible and have similar implementations in certain neuromorphic hardware.

**Weaknesses:**

1. The description in lines 94-107 is difficult to understand. Table 1 indeed shows that the "permuted" model performs better, but how does this relate to the previously acknowledged yet erroneous assumption that "the precise timing of the spikes should not affect the performance of the SNN"? There is no explanation of what the "permuted" operation is or how it relates to "the precise timing of the spikes." Additionally, why is the phenomenon named "temporal error information"? Where does the error manifest? Section 3.1 addresses these questions, but lines 94-107 should also provide some insight for readers encountering this for the first time.

2. I find the experimental work convincing; however, the description of the phenomenon of "temporal error information" needs further clarification. The current description leaves me unclear about what constitutes "temporal error information" in the original spike sequence. Which time steps contain erroneous spikes? What portion of the temporal errors is addressed by the "permuted" operation, and how does this improve the accuracy of the conversion?

**Questions:**

1. I am unclear about the details of the "permuted" operation. Figure 2(a) mentions that "the second Spiking phase outputs the same spike trains, but permuted." Since the output spike sequences are the same, where does the permutation manifest? Is it the firing times that have been rearranged?

2. In ANN-SNN conversion, even if the spike firing rates match the ANN activation values, the conversion is not lossless. Some works have discussed this, noting that in early time steps, there can be spikes that should not have been fired, resulting in erroneous spikes [1]. Is this related to the "temporal misinformation" mentioned in the paper? Furthermore, uneven errors also point to this issue [2]. What is the relationship between "temporal misinformation" and uneven errors? Could the authors provide some discussion on this aspect?

[1] X. He, Y. Li, D. Zhao, Q. Kong, and Y. Zeng, “Msat: biologically inspired multistage adaptive threshold for conversion of spiking neural networks,” Neural Computing and Applications, pp. 1–17, 2024.

[2] Tong Bu, Wei Fang, Jianhao Ding, PengLin Dai, Zhaofei Y u, and Tiejun Huang. Optimal ANN-SNN
conversion for high-accuracy and ultra-low-latency spiking neural networks. In International
Conference on Learning Representations, 2022c.

---

> ### Author Response · Authors · 2024-11-19
>
> **Weakness 1.** We hope that our common answer addresses the details of what we mean by a temporal misinformation and the reasons why we named it like that. We will update the lines in the introduction so that they are more informative and clear in this aspect.
>
> **Weakness 2.** We argue that the source of temporal misinformation is the instable firing rate in the early time steps of the baseline SNN models.
>
> SNN models converted from a pretrained ANN aim to approximate the ANN activation values with firing rates. The approximation is given by $\mathbf{x}^{(\ell)} \approx \bar{\mathbf{s}}^{(\ell)} = \frac{1}{T} \sum_{t=0}^{T} \mathbf{s}^{(\ell)}(t)$. However, in lower time steps, the approximation is too coarse as we can only use few spikes in order to approximate the ANN (continuous) values. For example, in $T=1$, the baselines are attempting to approximate ANN activations with binary values $0$ and $\theta$.
>
> What is crucial is that, at each spiking layer, the spiking neurons at early time steps, use only the outputs of the previous spiking layer from the same, early, time steps. As this information is already too coarse, **the approximation error accumulates throughout the network**, finally yielding in models that are underperforming in low latencies.
>
> Finally, with longer latencies, the model is using more spikes and is able to approximate the ANN values more accurately, and to correct the results from the first time steps. (One can argue that at long enough latencies, all reasonable conversion methods work equally well.)
>
> How do permutations fix this common occurrence?
>
> When performing permutations on spike trains after a spiking layer in the baseline models, the input to the next spiking layer in lower time steps **no longer depends only on the outputs of the previous layer in the same, lower time steps, but it depends on the outputs in all time steps $T$**.
> In particular, when a spiking layer is producing spike at time step $t=1$, it does so "taking into account" (via permutation) outputs at all the time steps from the previous spiking layer.
>
> As a way of example, consider two connected spiking neurons $N_1$ and $N_2$, where $N_1$ is sending the weighted output to $N_2$. If a spiking neuron $N_1$ in one layer has produced spike train $s = [1,0,0,0]$, in approximating ANN value of $.25$, then a spiking neuron $N_2$ at the first time step will use 1 as the approximation and will receive the input $W\cdot 1$ from neuron $N_1$. However, after a generic permutation of $s$, the probability of having zero at the first time step of output of neuron $N_1$ is $\frac{3}{4}$ (as oppose to having 1 with probability $\frac{1}{4}$), and at the first time step neuron $N_2$ will most likely receive the input $W\cdot 0=0$ from neuron $N_1$, which is a rather better approximation for $W\cdot .25$ than $W \cdot 1$.
>
> This property of receiving input at lower $t$ but taking into account the spike outputs of the previous layer at all the time steps is not only exclusive to lower $t$. Indeed, at every time step $t\leq T$, the input at a spiking layer is formed by taking into account spiking train outputs from the previous layer at all the time steps, but having already accounted for for the observed input at the first $t-1$ steps.
>
> In general, the permutations overall increase the performance of the baselines because the spike trains are "uniformized" in accordance to their rate, and the accumulation error is reduced. If a layer $l$ has produced spike outputs that well approximate the $l$ layer in ANN, then, after a generic permutation, at each time step starting with the first, the next layer is receiving the most likely binary approximation of those rates.
>
> This is nothing but Theorem 2 of the paper in visible action.
>
> We provide several evidence for our claims (new appendices) which are as follows: Appendix F observes what is the accuracy of the model at $t=1$, when we applied permutations on spike trains of higher length $T$; Appendix G shows the membrane potential distribution before firing in the baselines and permuted and TPP models. Permuted and TPP models show higher variance, hence their enhancement of the model's ability to produce spikes in low latency. Appendix G, which shows the experiments when particular classes of permutations are applied on the baseline models. In particular, the permutation that anchor first time step show the least increment in performance.

---

> > ### Comment · Reviewer_egA6 · 2024-11-20
> >
> > Thank you for your response. I find the additional explanation of "**Temporal information**" and "**Temporal misinformation**" to be clear.
> >
> > Regarding the source of misinformation, I agree with the authors' view that it stems from the "unstable firing rate in the early time steps of the baseline SNN models," which aligns with my own thoughts. The "permutation" operation ensures that at lower time steps, the input to the spiking layer no longer depends solely on the output of the previous layer at the same lower time steps, but rather on the outputs at all time steps, which also explains how permutation helps resolve temporal errors. The second point I mentioned in the weaknesses has been well addressed.
> >
> > I reviewed Appendix F, where at t=1, the permutation demonstrated a significantly higher accuracy compared to QCFS, which validates the advantage of "depending on the outputs at all time steps."

---

> ### Author Response · Authors · 2024-11-19
>
> **Q1.** You are absolutely correct. We rearrange the firing times according to the permutation of time steps.
>
> **Q2.** In addition to what we said about the instable firing rate of the baseline models in lower latency, and how permutations overcome this, we try to address how they potentially cope with the errors identified in [1] and [2]. For the SIN (spikes of innactive neurons) errors in reference [1], we recall that the SIN refers to occurrence when a spiking neuron emits a spike, even though the corresponding ANN pre-activation was negative. SIN can happen if some neurons in the previous spiking layer $(l-1)$ emitted spike trains modulated with positive weights $w_{+}^{l}$, before some other neurons in $(l-1)$ emitted spike trains modulated with negative weights $w^{l}_{-}$, which would eventually annulate the positive spike trains. In other words, we have two types of spike trains, one where the spikes are clustered early in time, and the other, where the spikes are clustered later on, so that the two spike streams do not integrate together in order to produce the correct input. Of course, this is a rather vast oversimplification of a potential situation, but it serves a point to show how permutations may overcome this situation.
>
> In fact, the unevenness errors in [2], describe a similar situation, where we have two stream of spikes, one where the spikes are clustered in the earlier and the other where the spikes are clustered in the later time steps. When combined together, the spike streams should produce the correct input, but due to their temporal misalignment, the subsequent neurons end up firing extra spikes or underfiring in general.
>
> Permutations, in general, tend to break clustering of spikes. In a simplified situation, if we have $m$ spikes in the course of $T$ time steps, the probability after a permutation of $m$ spikes being clustered together in $m$ time steps is $\frac{(T-m+1)\cdot m!}{T!}$. (Permutations have the similar effect on "voids", long subintervals in the spike trains without spikes). Overall, breaking clustering contributes to the uniformity of the input, and overall, elimination of  said errors.
>
> We also note that our answer here pertains to permutations in baseline models. For the TPP neurons, we kindly point your attention to the conversion error analysis in new Appendix I.

---

> > ### Comment · Reviewer_egA6 · 2024-11-20
> >
> > Thank you for the discussion on the errors mentioned in the two referenced papers. Although it doesn't fully correspond—since the existing early clustering and late clustering of spike firing is a simplified potential situation—this discussion helps illustrate how permutation can overcome such cases. It would be even better if the related discussion (the connection to other works) could be added to the revised version of the paper.
> >
> >
> > The authors have addressed and resolved my concerns well, and I will increase my rating by one point.

---

> > > ### Author Response · Authors · 2024-11-20
> > >
> > > Thank you once again for your insightful comments and for increasing the score! As per your suggestion, we will update the paper with connections to other works dealing with different types of errors in ANN-SNN setting.
> > >
> > > Authors

---

### Author Response · Authors · 2024-11-19
**Common answer to all the reviewers**

Thank you very much for your time in assessing our paper, and especially to reviewers _egA6_ and _gkwM_ for their valuable comments and questions. Here, we assess a common point for some of the reviews and try to clarify the notion of _temporal misinformation_.

**Permutation of a spike train:** By a permutation of a one-dimensional spike train $s$, we mean permutation of the entries of $s$. In other words, we rearrange the spike timings according to the permutation. One can refer to Figure 2 in the paper for the visual explanation.


+ **Temporal information**: If permuting spike trains after spike layers degrades the performance of the SNN, it means that the _positioning_ of the spikes carries **useful** information for the network and its performance. Hence, we would call this occurrence **"Temporal information"** in the spike trains/networks.
+ **Temporal misinformation**: In the opposite situation, when the performance of the network increases when permuting spike trains, it means that the spiking layers produce spike trains with timings which are "misinformative", and we call this occurrence **"Temporal misinformation"**. Our article studies temporal misinformation in ANN-SNN conversion setting.


**Name "Temporal misinformation":** By a temporal misinformation we mean a phenomenon in rate encoded SNNs when permuting spike trains after spiking layers of the model increases the performance of the model (in the expectation). Permutations are done in orderly fashion: the output spike trains of one layer are permuted and then passed to the next spiking layer for processing, whose output spike trains are then permuted and passed to the following spiking layer and so on.

**New experiments in the APPENDIX:** As few reviewers commented on the source of temporal misinformation, we provided our point of view on this, and extended experiments to support it. Our claim is that the baseline SNN models suffer from unstable firing rates in the early spike times, due to the their inability to approximate the ANN values with few time steps.

In the Appendices F, G, and H we provided new experiments concerning permutation effect of SNN baselines:


+ In Appendix F we show how the permutations affect the early time steps in the output layer, explaining why the performance of the baseline drastically increases when permutations are applied. This is Theorem 2 of the paper in visible action.
+ In Appendix G, we plot membrane potential distributions before firing of the baselines and contrast them with the distributions of permuted and TPP models. These plots show how permutations and TPP neurons provide wider distributions, especially so in lower time steps, increasing the networks variability and spiking activity.
+ In Appendix H, we provide new experimental results concerning the effect of permutations in some particular cases.
    - Figure 10: We plot the accuracies of the model when all of $T!$ ($T=4$) different permutations are applied. In particular, we notice that the permutations anchoring the first time step, show the least effect, confirming our claim above.
    - Figure 11: We plot the accuracies of the model when various classes (28) of them, of permutations of 8 elements ($T=8$) are applied, fixing two time steps. The figure shows one again, that the permutations fixing the first element (in combination with any other) show lower performance increase as compared to the variations.
    - Figure 12 and 13 show the effect of a permutation when it is applied only on a single layer in the baseline model. We notice that the permutation show the least effect when they are applied in the early layers (as the SNN model is still approximating the ANN well), and in the late layers (as it is already too late to fix the accumulated approximation error from the previous layers).

Finally, in the Appendix I, we provide a new theoretical result concerning TPP setting, which provides further insights into conversion error made when using them in ANN-SNN conversion.

We hope that these new results shed more light on our method, and confirm its potential in ANN-SNN conversion.

---

### Meta-Review · Area_Chair_1gjB · 2024-12-20

**Metareview:**

This paper presents a new form of spiking neural network (SNN) unit to be used for ANN-to-SNN conversion. Specifically, the authors begin by examining the importance of precise spike times in an SNN following conversion. They observe a surprising phenomenon which they dub “temporal misinformation”, whereby the spike times do matter but in the opposite way to what a naive observer may have predicted. Namely, if the spike times are permuted the performance *increases* rather than decreasing, showing the spike times matter, but the spike times obtained following conversion may not be optimal. Inspired by this, the authors develop a two-stage spiking neuron model that uses a phase of integration followed by a phase of stochastic spike generation. They claim that their SNN model provides better performance following ANN-to-SNN conversion, and they support this claim with data on a series of classic image categorization datasets.

The strengths of this paper are that it uncovers an interesting, surprising phenomenon in ANN-to-SNN conversion models (temporal misinformation) and uses this observation to improve on current techniques. The weaknesses are (1) the reasons for this interesting phenomenon and the success of the new technique are not thoroughly explored leaving the reader feeling unsure what is really happening, and (2) the paper’s presentation is not great, with the writing and methods very hard to follow (e.g. the specific contributions were not originally well described, and the description of the conversion process in Section 3 and Appendix A do not provide critical pieces of information for understanding).

On the first point, the authors provided some additional experiments in rebuttal that helped, though could be greatly expanded. The second point remained largely unaddressed. Indeed, on the second point several of the reviewers seemed perplexed by the paper, and none of them gave it a good rating on presentation (even the positive reviewers). In the end, the final scores left this paper as a borderline case, and the AC had to make a judgement call. Based on the considerations above, a decision to reject was reached. It is the AC’s assessment that there is something potentially very interesting in this paper, but the authors should revise the paper significantly and submit elsewhere, so that they have the time to properly focus on explaining and understanding “temporal misinformation” and providing more grounded explanations for their technique. As well, the AC encourages the authors to generally review the clarity of the paper before resubmitting elsewhere.

**Additional Comments On Reviewer Discussion:**

The discussions during the rebuttal for this paper were not great. It is the AC’s opinion that some of the reviewers were making unconstructive comments about novelty and typos, something the authors highlighted to the AC, and with which the AC agrees. But, at the same time, the AC feels that authors took an aggressive approach in the rebuttals and failed to help the reviewers understand their paper better. It was clear from the reviews that there was an issue of clarity and presentation, and yet the authors seemed resistant to addressing this. In discussion between the AC and reviewers, even the positive reviewers noted that the paper was hard to follow and that the authors had not addressed these issues of clarity in a constructive manner with the other reviewers. In the end, the AC decided not to consider the points around novelty and typos, but did significantly factor in general issues of presentation when coming to their decision.

---

### Decision · Program_Chairs · 2025-01-22

Reject